# AutoManual: Constructing Instruction Manuals by LLM Agents via Interactive Environmental Learning

**Minghao Chen**[1], **Yihang Li**[2], **Yanting Yang**[3], **Shiyu Yu**[5], **Binbin Lin**[3,4*], **Xiaofei He**[2]

[1]School of Computer Science, Hangzhou Dianzi University
[2]State Key Lab of CAD&CG, Zhejiang University
[3]School of Software Technology, Zhejiang University  [4]Fullong Inc.  [5]NingBo Port Group
minghaochen01@gmail.com

## Abstract

Large Language Models (LLM) based agents have shown promise in autonomously completing tasks across various domains, e.g., robotics, games, and web navigation. However, these agents typically require elaborate design and expert prompts to solve tasks in specific domains, which limits their adaptability. We introduce AutoManual, a framework enabling LLM agents to autonomously build their understanding through interaction and adapt to new environments. AutoManual categorizes environmental knowledge into diverse rules and optimizes them in an online fashion by two agents: 1) The Planner codes actionable plans based on current rules for interacting with the environment. 2) The Builder updates the rules through a well-structured rule system that facilitates online rule management and essential detail retention. To mitigate hallucinations in managing rules, we introduce a *case-conditioned prompting* strategy for the Builder. Finally, the Formulator agent compiles these rules into a comprehensive manual. The self-generated manual can not only improve the adaptability but also guide the planning of smaller LLMs while being human-readable. Given only one simple demonstration, AutoManual significantly improves task success rates, achieving 97.4% with GPT-4-turbo and 86.2% with GPT-3.5-turbo on ALFWorld benchmark tasks. The code is available at https://github.com/minghchen/automanual.

## 1 Introduction

Recently, autonomous agents based on Large Language Models (LLM), e.g., ReAct [33], Reflexion [16], SayCan [1], WebGPT [10], and Voyager [24], have demonstrated their potential to complete long-horizon tasks in grounded environments. These LLM agents operate by generating thoughts and actions that are executable in the environment. For customized environments, such as robotics [1, 7, 18, 23] and games [14, 24, 39], prior methods provide detailed instructions and in-context examples to familiarize LLM with action functions (API) and the target environment. However, unlike these agents, humans can autonomously build and update their understanding of an unfamiliar environment through dynamic interaction.

Several existing methods enable LLM agents to reflect on feedback [16, 21] or save successful experiences as skills [21, 24, 35] to enhance the performance and reduce the reliance on human-provided examples. However, these reflections and skills have not been well exploited to foster a deeper understanding of the environment. As a result, directly using saved skills as in-context examples can lead to the **Path Dependence** problem, i.e., the agent blindly replicates the paths of previous successes, failing to adapt appropriately to new scenarios. Such problems are more severe in real-world situations characterized by high variability.

38th Conference on Neural Information Processing Systems (NeurIPS 2024).

A previous work, ExpeL [35], gathers the trajectories of LLM agents and extracts cross-task rules from them. However, these rules are extracted offline, making ExpeL suffer from the same distributional shift problem as Offline RL [6]. Meanwhile, due to the simplicity of rule management, its rules are always armchair general and unhelpful for the Path Dependency problem. In this paper, we propose a novel framework called **AutoManual** to build a well-organized understanding of the environment that can guide multi-task planning effectively. AutoManual leverages a dynamic rule system that not only extracts valuable experience, including skills and reflections, into different types of rules but also allows for continuously updating these rules in response to new situations. Additionally, error-prone details are explicitly described in the rules to improve the robustness of planning.

AutoManual follows two alternating iterative processes to optimize the rules. First, given the observation and task of an episode, the Planner agent utilizes currently discovered rules to write free-form code as an actionable plan. The interaction between the environment and the Planner will loop until the episode ends. Second, based on this trajectory, the Builder agent will update relevant rules through the rule system. This online updating mechanism can timely verify whether the rules have deviations and are applicable to the Planner. After rules optimization, the Formulator agent categorizes these rules according to their application scenarios and compiles a comprehensive manual in Markdown format.

The challenge lies in enabling the Builder to accurately extract applicable rules from a long trajectory, as LLM are prone to generating hallucinations. To address this, we employ a *case-conditioned prompting* strategy, which directs the Builder to focus on specific rules according to the case of the trajectory. For example, if errors occurred in a trajectory, the Builder is first asked to determine which caused the error: an unrecorded situation occurred, or the Planner failed to follow existing rules. Based on this answer, the Builder will be given corresponding prompts to update relevant rules.

To summarize, our contributions are the following:

- We adopt actionable code as the way for the Planner agent to interact with the environment. We introduce a structured rule system that allows the Builder agent to manage multiple types of knowledge from these code-based interactions.

- We propose an alternating process between the Planner and Builder agents to optimize rules in an online manner and resolve the Path Dependency problem. To improve readability, the Formulator agent is introduced to reorganize and formalize the rules into a Markdown manual.

- To facilitate rule management, we employ a *case-conditioned prompting* strategy, which guides the Builder to manage specific types of rules for different trajectory cases.

- Starting from a single demonstration, AutoManual can generate detailed instruction manuals for complex environments like ALFWorld and MiniWoB++. These manuals allow LLM agents to achieve remarkable success rates of 97.4% with GPT-4-turbo and 86.2% with GPT-3.5-turbo on ALFWorld, 98.3% with GPT-4-turbo and 92.7% with GPT-3.5-turbo on MiniWoB++.

## 2  Related Works

### 2.1  LLM for Agents Planning

Large Language Models (LLM) exhibit powerful reasoning and planning capabilities [11, 12, 28, 33, 39] while requiring much fewer demonstrations than traditional learning methods. With this planning capability as the core, LLM agents are being developed for use in robotics [1, 7, 18, 20, 23], game-playing [14, 24, 27, 39], software development [3, 15], and other fields [30]. Prior studies [16, 21, 33] allow agents to adjust actions or plans based on environmental feedback to improve planning performance. Given the powerful programming capability of LLM, several works, e.g., CodeAsPolicy [7], ProgPrompt [18] and AdaPlanner [21], propose to use Python code as the plan of LLM agents. This form of output can automatically respond to in-plan feedback and achieve better performance than the action and JSON format [21, 25].

### 2.2  Self-improvement of LLM Agents

Embodied agent research has long sought to enable agents to self-improve through interactive experiences. Unlike traditional learning-based agents that require extensive iterations for optimization,

Reflexion [16] allows LLM agents to reflect on previous failures and quickly improve their plans. Some works [32, 34, 36] combine tree search with reflection to deliberately seek a better solution. Apart from failure experiences, prior studies [21, 24, 39] utilize successful experiences as skills to assist future planning. Voyager [24] stores generated and verified programs into the skill library as a new skill for more complex tasks. AdaPlanner [21] also discovers and archives successful programs into skill memory for future similar tasks. However, these methods stop updating skills after storing them, which inevitably leads to the Path Dependency problem.

Another series of works [26, 31, 38] employs LLM as a prompt optimizer to enhance its own performance. In contrast to our approach, which addresses challenges in unfamiliar environments, these studies focus on enhancing LLM reasoning performance. As a result, their optimized prompts are typically brief and lack environmental knowledge.

### 2.3 Memory Management of LLM Agents

For LLM agents, learning from past experiences can also be viewed as managing the episodic memory [16]. CLIN [9] proposes to keep updating a memory centered on causal abstractions for new trials. Retrieval-Augmented Planning (RAP) [4] retrieves past experiences corresponding to the current situation. MemGPT [13] allows LLM to select content to retain in working memory and to search for information in long-term memory. Generative Agents [14] retrieve memories based on recency, importance, and relevance to the current situation. Generative Agents also generate tree-structured reflections, but they focus on a continuous scenario rather than task-oriented rules.

### 2.4 LLM for Rule Discovery

Several recent works also investigate the rule discovery capabilities of LLM. Zhu et al. [40] propose Hypotheses-to-Theories (HtT), enabling LLM to induce and deduce rules for basic reasoning tasks. For LLM agents, ExpeL [35] gathers the trajectories of Reflexion agents and extracts cross-task rules from them. Furthermore, AutoGuide [2] generates state-aware rules and retrieves rules relevant to the test-time state. Unlike ExpeL and AutoGuide, which extract rules from offline experiences, we update rules in an online manner, verifying their reliability and applicability. For more discussion of differences, refer to Appendix C.

## 3 Methods

### 3.1 AutoManual Overview

Our AutoManual framework, shown in Fig 1, consists of three main stages. **Building stage:** The Planner agent and Builder agent collaborate to build rules from the interactive environment. The Consolidator agent merges or deletes redundant rules when the rules exceed the maximum rule number. **Formulating stage:** The Formulator agent categorizes the rules, summarizes the key points, and formulates them into a manual in Markdown form. **Testing stage:** Based on the generated manual, a test-time Planner agent will be evaluated through test tasks and scenarios.

Formally, an Interactive Environment can be modeled as a Partially Observable Markov Decision Process (POMDP): $(\mathcal{S}, \mathcal{A}, \mathcal{T}, \mathcal{G}, \mathcal{O})$. At the start of each episode, a scenario $s_0 \in \mathcal{S}$ will be initialized, a text-grounded task $g \in \mathcal{G}$ and the initial observation $o_0 \in \mathcal{O}$ (processed into textual form) will be given. The environment can be interacted with through permissible actions (API) set $\mathcal{A}$. After executing an action $a \in \mathcal{A}$, the environment will return the result of the action and the new observation $o'$ based on the dynamics $T(s'|s, a) \in \mathcal{T}$ and $O(o'|s')$. Finally, when the episode is done, a binary reward $r \in \{-1, 1\}$ indicating the failure or success of the task will be returned.

We approach the learning of environmental rules as an optimization problem:

$$\max_{\Theta} E_{s_0, g} E_{\rho(\cdot|\Theta)} r(\tau_\rho) \tag{1}$$

where $\Theta$ denotes all rules in our rule system, $\rho(\cdot|\Theta)$ denotes the policy of the Planner given the current rules $\Theta$ and $\tau_\rho$ denotes a trajectory of $\rho(\cdot|\Theta)$ starting from $[o_0, g]$. Classic policy gradient methods [29] solve such problems through stochastic gradient ascent, i.e., executing the current policy to obtain the episodic reward and back-propagating gradients to update the parameters.

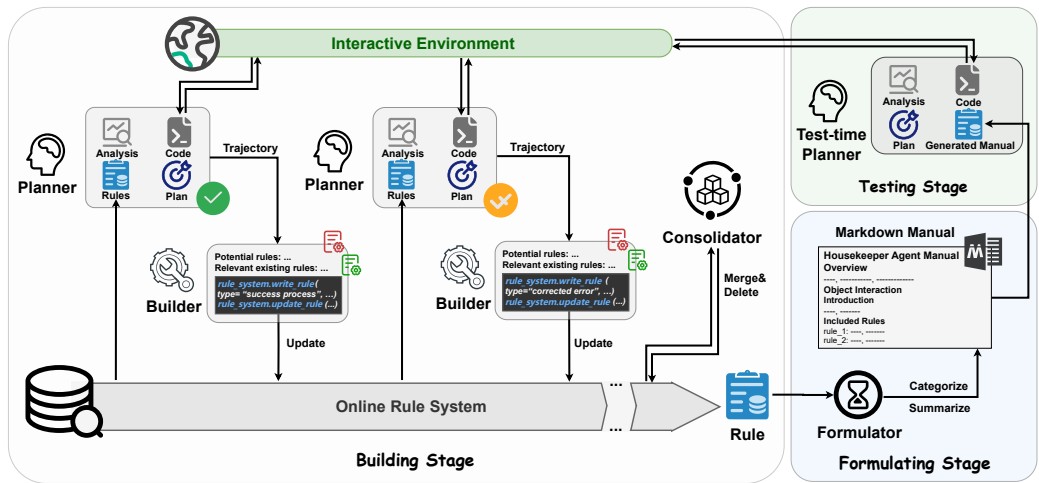

Figure 1: **AutoManual Overview:** AutoManual operates in three stages: (1) **Building Stage:** The Planner agent interacts with the environment by coding actionable plans. After receiving the current trajectory of the Planner, the Builder agent manages rules through the online rule system. (2) **Formulating Stage:** The Formulator agent formulates the resulting rules into a Markdown manual. (3) **Testing Stage:** A test-time Planner agent utilizes the manual to complete testing tasks.

Inspired by this online reinforcement learning paradigm, we follow two alternative processes to optimize the rules $\Theta$: 1. The Planner practices the current rules through interaction within an episode. 2. The Builder updates the rules $\Theta$ based on this trajectory. Compared to traditional parameter optimization, sample-inefficient gradient ascent is replaced by text-based rule management. We design a well-structured rule system described in Section 3.3 to ensure the rule updating contributes to rewards. Additionally, to limit the role of human expertise, we only provide a simple example demonstrating the output format to agents. Then, manually derive several initial rules from this example as the starting point of the optimization.

## 3.2 Planner Agent for Interactive Planning

As demonstrated by the success of Voyager [24] and AdaPlanner [21], code-based planning can leverage the powerful programming capability of LLM and automatically react to in-plan feedback. Voyager and AdaPlanner output and refine a complete solution function for the task, which is potentially reusable. However, this function-form output is difficult to adjust in response to environmental feedback, as it requires maintaining the integrity of the plan throughout.

Our Planner Agent outputs free-form code as its plan, which aligns more with the natural programming capabilities of LLM [7, 22]. This form simplifies planning by only generating code necessary for the current environmental situation and feedback without the overhead of integrating previously executed code. As shown in Fig 2, at the start of a new episode, the Planner receives system prompts, current rules $\Theta$, relevant samples from the skill and reflection libraries, the target task $g$, and initial observation $o_0$. System prompts contain the role, permissible actions $\mathcal{A}$, response guidelines, and a simple example (detailed in Appendix H). The output of the Planner is structured into four segments during each cycle:

1. **Analysis:** The understanding of the current situation and reflection on previous errors if exist.

2. **Related Rules:** Rules (along with their IDs) that need to be considered in this situation.

3. **Overall Plan:** The general plan to complete the task.

4. **Code:** A block of Python code divided into steps. The Planner is encouraged to define helpful functions in the code, which might be reusable in similar scenarios.

We denote this response of the Planner as $[thought_t, code_t]$, where $thought_t$ denotes the first three segments. $code_t$ executed in the environment is followed by feedback $c_t$, which informs the subsequent output cycle. This process iterates until the episode ends or a response limit is reached.

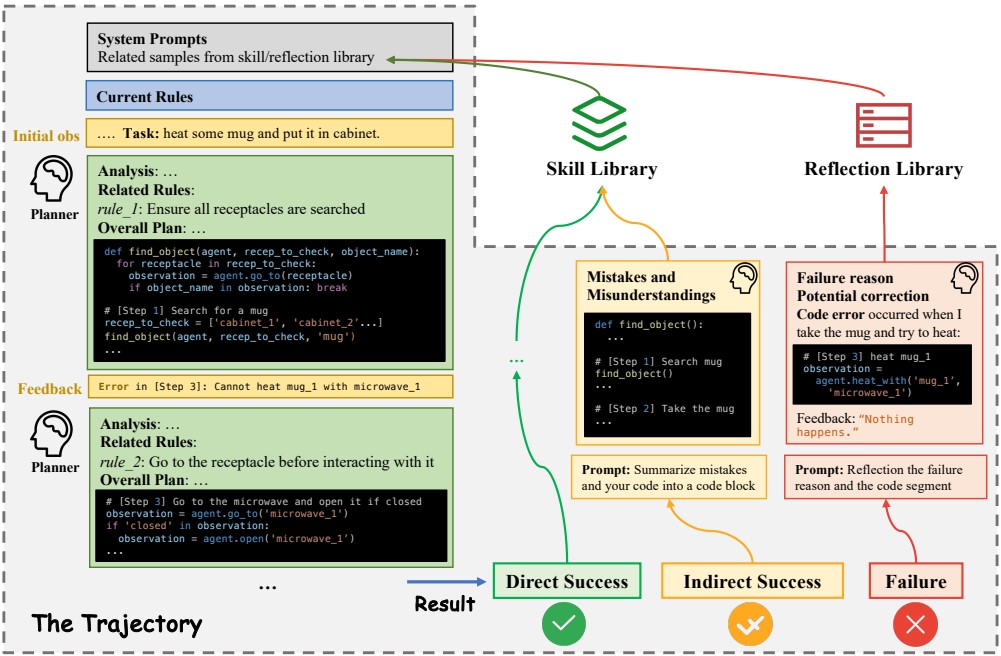

Figure 2: **The Planner Trajectory:** Given the current task and rules, the Planner will interact with the environment through free-form code. Based on the trajectory result, the Planner will generate a corresponding conclusion, which will be saved in the skill or reflection library.

As shown in Fig 2, according to the episodic reward, we categorize the result into **Direct Success**, **Indirect Success** (errors occurred but were solved later), and **Failure**. In the case of Direct or Indirect Success, the Planner will be prompted to organize its previous code into a code block. For Indirect Success, it additionally summarizes the mistakes and misunderstandings that cause errors. For the Failure case, the Planner will be prompted to reflect on the reason for the failure carefully, suggest reasonable corrections, and specify the code segment that caused the error. We denote this response of the Planner as *conclusion*. Finally, we obtain a trajectory of the Planner:

$$\tau_\rho = (o_0, g, [thought_1, code_1], c_1, ..., [thought_T, code_T], c_T, conclusion) \tag{2}$$

**Skill Library and Reflection Library:** Apart from rules, we also manage and transmit conclusions from previous episodes, which provide essential details for generating planning code. In the case of Direct or Indirect Success, we save the code block in *conclusion* as a skill for that task type [1] into the skill library [21, 24]. In the Failure case, we save its *conclusion* as a reflection for that task type into the reflection library. When a new task comes, the code block of the most similar task is retrieved from the skill library. If there is no existing skill for the new task type, the reflection for that task type will be returned. As mentioned in the Introduction, compared with rules, these skills and reflections contain more programming details but are less generalizable to new scenarios, i.e., the Path Dependence problem.

**Cooperation between Agents:** In our framework, rule management is not solely the responsibility of the Builder; the Planner also plays a critical role by explicitly identifying the rules it engages in its response. This cooperation is facilitated by including the Planner's thoughts within the trajectory $\tau$, which is provided to the Builder. This synergy enhances the identification and adjustment of problematic rules. In addition, *conclusion* from the Planner contains the detailed success process or reflections on errors, which further assist the Builder in managing corresponding types of rules.

### 3.3 Builder and Consolidator Agents for Rule Management

Upon receiving the trajectory $\tau_\rho$, the Builder has to manage the rules through the rule system.

---

[1]All tasks in ALFWorld are divided into 6 task types, e.g., pick_heat_then_place, look_at_obj_in_light. For each task type, we store only one skill code.

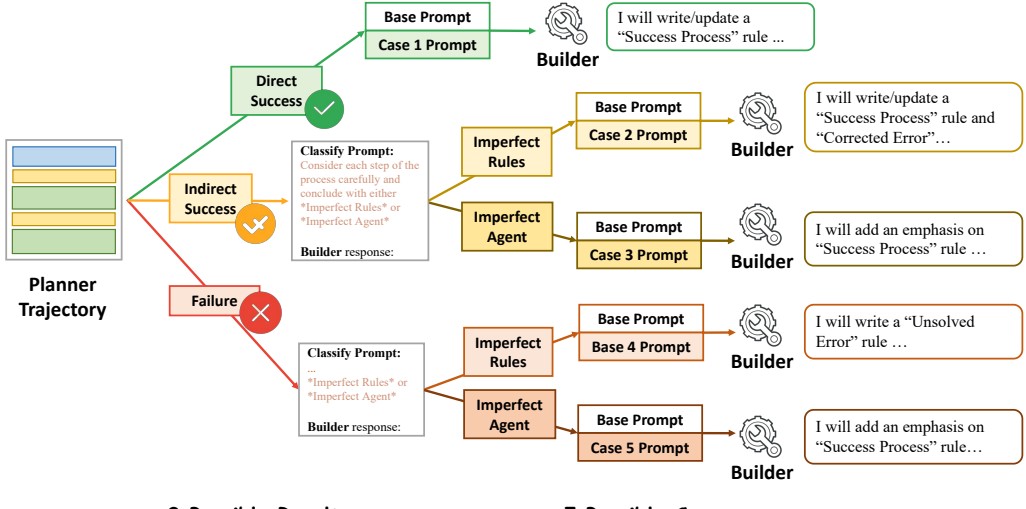

Figure 3: **Case-Conditioned Prompts:** Given the current trajectory, the Builder classifies the cause of the major error as "Imperfect Rules" or "Imperfect Agents". Then, the Builder will get the base prompt and corresponding prompt to guide its rule management.

**Rule System:** We intuitively identify rules as the kinds of knowledge that help task completion, including the analyses of the observed phenomenon $T(o'|o, a)$, the mechanism $T(s'|s, a)$, and the correlation between the reward $r$ and $\tau_\rho$, i.e., the success process or the occurred error. Therefore, unlike ExpeL [35] and AutoGuide [2], which derive general insight from the trajectory, our system categorizes six specific rule types to extract environmental knowledge that targets different aspects of the trajectory. Furthermore, each rule in our system is enhanced with **Example** attribute to illustrate its application and important details, making it grounded and well-understood. Specifically, each rule in the rule system has these four attributes:

1. **Rule Type:** The type of the rule, chosen from ["Special Phenomenon", "Special Mechanism", "Useful Helper Method", "Success Process", "Corrected Error", "Unsolved Error"];

2. **Rule Content**: A description of the rule, beginning with the scope of its applicable scenarios;

3. **Example**: An example or code from the trajectory demonstrates this rule, where additional remarks, e.g. error-prone details, can also be added to it;

4. **Validation Logs**: Logs that track the rule's application and updates, including episode and rule IDs that trace the rule's evolution, serving as a reference for the Builder and Consolidator.

The Builder manages the rules through the following functions of the rule system:

- *write_rule(\*\*rule_attributes)*: Write down a new rule with its four attributes.
- *update_rule(rule_id, \*\*rule_attributes)*: Rewrite the attributes of a existing rule.
- *stop_generating()*: When the trajectory is not needed or insufficient to derive any more new rules, the function should be called.

Similar to hierarchical reflections in Generative Agents [14], we allow the Builder to utilize existing rules to induce more general or deeper rules and record their dependence in Rule Content or Validation Logs, more discussed in Appendix D.

**Case-Conditioned Prompting:** To mitigate the risk of erroneous rule creation, such as deriving rules of success from a failed trajectory, we employ case-conditioned prompts. As illustrated in Fig 3, the Builder first analyzes and determines if the major errors stem from "Imperfect Rules" or "Imperfect Agent". Based on this analysis and the trajectory results, targeted prompts guide the Builder in rule management [2]. For example, in a case of indirect success due to imperfect rules (Case 2), the prompts will guide the Builder to extract or update the success process, helper methods, and error reflections

---

[2]**Notice**: These prompts for the Builder are environment-independent and shared across all environments.

in corresponding rule types. Finally, the Builder responds with the potential rules detailing their relation with existing rules and uses the functions of the rule system to manage rules.

**Rule Consolidation:** When the number of rules in the rule system exceeds $N_{max}$, the Consolidator agent steps in to consolidate related rules and delete redundant rules. It uses three functions of the rule system: *get_trajectory(episode_id)*, *update_rule(rule_id, \*\*rule_attributes)* and *delete_rule(rule_id)*. Given the current rules, the Consolidator identifies potentially relevant or overlapped rules, uses *get_trajectory* function to investigate the trajectories they depend on, and finally calls the remaining functions to manage the rules. During the management, the Consolidator ensures that consolidation retains details of rules and examples.

### 3.4 Manual Formulation

Once the building stage is complete, we can obtain a set of rules targeted to different situations, whose applicability has been validated through online optimization. Our next goal is to enhance their readability and global understanding. To achieve this, we introduce the Formulator agent, designed to transform these rules into a user-friendly manual, analogous to a teacher imparting a wealth of knowledge through easily digestible lessons. As depicted in Fig 1, the Formulator begins by categorizing all rules based on their target scenarios. This categorization aids in structuring the manual and ensures that related rules are discussed together, which enhances the logical flow and accessibility of the information. For each category, the Formulator drafts an introduction, summarizing the rules it contains and highlighting the key points and overall principles that govern the specific scenarios. Finally, the Formulator compiles the rules and their introductions into a comprehensive manual formatted in Markdown.

## 4 Experiments

In line with prior works [2, 21], we conduct the experiments on three interactive environments: (1) **ALFWorld** [17] is a text-based virtual household environment containing six distinct task types. We run the building stage on 36 tasks (6 tasks for each task type) sampled from the training set of ALFWorld, and each task is run only once. Following previous works [16, 21, 33], we run the testing stage on the validation unseen set containing 134 tasks across these six types. (2) **MiniWoB++** [8] is a simulated web environment where agents complete diverse tasks on the Internet by performing keyboard and mouse actions. Prior works [5, 21] selects 9 task types with environmental feedback and 44 task types without feedback from MiniWoB++ tasks. We perform experiments on 9 task types with feedback or on all 53 task types. At each stage, we randomly sample 6 tasks for each task type. (3) **WebArena** [37] introduces realistic web environments by emulating the functionality and data of popular websites. This benchmark poses significant challenges for LLM agents due to its large observation and action space, along with tasks that require longer planning horizons. Following AutoGuide [2], our experiments focus on the Reddit domain within WebArena.

During the building and formulating stages, we use GPT-4-turbo (`gpt-4-1106-preview`) for all agents. At the testing stage, we equip the Planner agent with GPT-4-turbo or GPT-3.5-turbo (`gpt-3.5-turbo-1106`), to evaluate the effect of generated manuals on relatively smaller LLM.

**Compared Methods:** In the experiments, we compare AutoManual with the following methods of LLM Agent: (1) **ReAct [33]** prompts LLM to generate the reasoning trace using CoT [28] and next-step action; (2) **Reflexion [16]** agents generate reflection on task feedback signals, which is saved in the memory for subsequent trials; (3) **ExpeL [35]** extract insights and skills from the offline trajectories of Reflexion agents; (4) **RCI [5]** agent recursively criticizes and improves its output for solving computer tasks; (5) **AdaPlanner [21]** allows the LLM agent to generate and adaptively refine a code-style plan; (6) **Planner+Lib.** represents our Planner agent equipped with skill and reflection libraries (§3.2) during building and testing stages without any rules. We re-implement prior methods with GPT-3.5 and GPT-4 versions the same as ours for fair comparisons.

ReAct, Reflexion, and ExpeL provide LLM agents with 12 human examples (2 examples per task type) of ALFWorld. For AdaPlanner, they provide 6 human examples (1 example per task type) of ALFWorld as the start of skill discovery. For our methods, agents are provided only one human example of the simplest task (`Put`) on ALFWorld. On MiniWob++, our agents are provided one human example (`search-engine`) for tasks with feedback and 4 examples for all tasks. On WebArena, our

Table 1: Success rate (%) of LLM agent methods on ALFWorld test tasks. For each method, the number of all human examples used is listed. "Planner+Lib." represents only using skill&reflection library during the building and testing stages. We run all experiments 3 times and show the average.

| Methods | Examples | Put | Clean | Heat | Cool | Examine | Put two | ALL |
|---|---|---|---|---|---|---|---|---|
| *Testing LLM: GPT-3.5-turbo* | | | | | | | | |
| ReAct [33] | 12 | 75.0 | 24.7 | 37.7 | 36.4 | 44.4 | 11.8 | 41.9 |
| Reflexion [16] | 12 | 87.5 | 44.1 | 73.9 | 50.0 | 61.1 | 35.3 | 59.8 |
| ExpeL [35] | 12 | 62.5 | 61.3 | 30.4 | 61.9 | 55.5 | 35.3 | 52.2 |
| AdaPlanner [21] | 6 | 83.3 | 46.2 | 65.2 | 74.2 | 68.5 | 52.9 | 63.3 |
| Planner+Lib. | 1 | 77.8 | **88.2** | 82.6 | 72.7 | 37.0 | 27.5 | 66.5 |
| AutoManual | 1 | **95.8** | 79.6 | **87.0** | **78.8** | **100.0** | **66.7** | **86.2** |
| *Testing LLM: GPT-4-turbo* | | | | | | | | |
| ReAct [33] | 12 | 95.8 | 76.3 | 69.6 | 86.4 | 72.2 | 52.9 | 76.8 |
| Reflexion [16] | 12 | **100.0** | 95.7 | 78.3 | 86.4 | 77.8 | 70.6 | 85.9 |
| ExpeL [35] | 12 | 94.4 | 82.8 | 72.4 | 81.8 | 72.2 | 58.8 | 79.2 |
| AdaPlanner [21] | 6 | 88.9 | 90.3 | 85.5 | 75.8 | 64.8 | 41.2 | 76.4 |
| Planner+Lib. | 1 | **100.0** | 93.5 | **100.0** | 93.9 | 88.9 | 39.2 | 88.1 |
| AutoManual | 1 | **100.0** | **98.9** | **100.0** | **95.4** | **100.0** | **90.2** | **97.4** |

Table 2: Success rate (%) of LLM agent methods on 9 task types with feedback and all 53 task types of MiniWoB++. For each method, the number of human examples used is listed.

| Methods | Examples | With feedback (9 types) | Examples | ALL (53 types) |
|---|---|---|---|---|
| *Testing LLM: GPT-3.5-turbo* | | | | |
| RCI [33] | 22 | 45.6 | 104 | 77.3 |
| AdaPlanner [21] | 13 | 71.6 | 38 | 89.4 |
| Planner+Lib. | 1 | 63.6 | 4 | 87.0 |
| AutoManual | 1 | **82.2** | 4 | **92.7** |
| *Testing LLM: GPT-4-turbo* | | | | |
| RCI [33] | 22 | 60.4 | 104 | 88.6 |
| AdaPlanner [21] | 13 | 74.1 | 38 | 90.3 |
| Planner+Lib. | 1 | 80.2 | 4 | 94.4 |
| AutoManual | 1 | **94.5** | 4 | **98.3** |

agents are also provided with one human demonstration. To reduce randomness, we performed each experiment three times and reported the average. More details of the implementation and prompts for AutoManual can be found in the Appendix.

## 4.1 Main Results

**Main Results on ALFWorld:** As shown in Tab. 1, AutoManual significantly outperforms the existing methods, evidenced by overall success rates of 86.2% when using GPT-3.5-turbo for the testing stage and 97.4% when using GPT-4-turbo. Noticeably, AutoManual requires little expert prior knowledge about the environment and is only provided with one human example to achieve excellent results. In comparison, the rules induced by ExpeL hardly improve performance, as its offline trajectories are composed of individual actions rather than code.

Table 3: Test on WebArena (Reddit).

| Methods | Examples | Suc(%) |
|---|---|---|
| ReAct [37] | 2 | 6.0 |
| AutoGuide [2] | 19 | 43.7 |
| SteP [19] | 14 | 55.0 |
| Planner | 1 | 51.1 |
| AutoManual | 1 | **65.1** |

We find the performance of AdaPlanner is lower than reported. One reason is that AdaPlanner requires LLM to output specific formats to complete its function-form code, which is difficult for creative LLM, e.g., GPT-4-turbo. In addition, AdaPlanner and Planner+Lib. are inferior to AutoManual because they only store successful paths as skills and inevitably face the Path Dependence problem. Especially, tasks in *Put Two* have various scenarios, such as "two objects can occur at the same receptacle or different receptacles", that require different processes to solve (Appendix G shows an example). Furthermore, Planner+Lib. often does not mark error-prone points in its skills, such as "target objects may appear in unconventional locations".

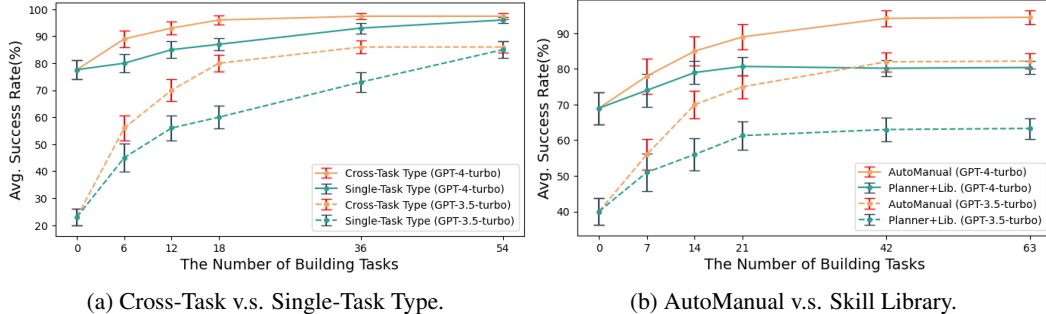

(a) Cross-Task v.s. Single-Task Type.  (b) AutoManual v.s. Skill Library.

Figure 4: (a) The success rate curve with standard deviation when testing GPT-4-turbo or GPT-3.5-turbo on ALFWorld. Building is performed cross-task or single-task type. (b) The success rate curve with standard deviation using AutoManual or Planner+Lib. when testing with GPT-4-turbo or GPT-3.5-turbo on 9 task types with feedback in MiniWob++.

Table 4: Ablation study of AutoManual on ALFWorld when testing with GPT-4-turbo.

| Online | Skill&Reflect Lib. | Case Prompt | Formulation | Avg. Error Steps ($\downarrow$) | Success Rate (%) |
|--------|--------------------|-------------|-------------|--------------------------------|------------------|
|        |                    |             |             | 2.3 | 77.6 |
|        | ✓                  |             |             | 1.5 | 88.1 |
|        | ✓                  | ✓           | ✓           | 1.3 | 90.7 |
| ✓      |                    | ✓           | ✓           | 1.6 | 89.5 |
| ✓      | ✓                  |             | ✓           | 1.0 | 93.8 |
| ✓      | ✓                  | ✓           |             | 0.5 | 96.5 |
| ✓      | ✓                  | ✓           | ✓           | **0.3** | **97.4** |

**Main Results on MiniWoB++:** As shown in Tab. 2, the performance of AutoManual exceeds the previous methods and Planner+Lib. by a large margin. Especially in 9 task types with feedback, these tasks have higher diversity and require LLM agents to cope with various situations. For example, the tasks in *login-user-popup* type will interrupt the agent's plan at any time, requiring the agent to cope with unexpected situations. Therefore, solely imitating previous successful experiences without extracting targeted rules will lead to task failure. Additionally, due to the flexibility of free-form codes, our method shows better adaptability while requiring fewer expert examples than prior methods.

**Learning Curves.** We show the success rate curves (testing with GPT-4-turbo or GPT-3.5-turbo) when gradually increasing the tasks of the building stage in Fig 4. In the left image, we share rules across all task types (Cross-task Type), as in AutoManual, or each task type builds a separate set of rules (Single-task Type) during the building stage. Fig 4 (a) demonstrates that sharing rules across task types can facilitate rule optimization. The rules for each task type deepen understanding of the environment, thereby boosting the planning of other tasks. In Fig 4 (b), we compare AutoManual and Planner+Lib. on 9 tasks with feedback in MiniWob++. We find that Planner+Lib. tends to get stuck with a lower success rate. In the face of highly diverse scenarios, Skill Library cannot express the rules behind the environment, thus falling into the Path Dependency problem.

## 4.2 Ablation Study

In this ablation study, we quantify the impact of each core component of the AutoManual framework on performance, specifically focusing on success rates and error reduction during task execution. Since we allowed the Planner to replan up to 3 times, each task could have up to 4 error steps.

**Online v.s. Offline Rule Management:** We perform offline AutoManual by collecting all trajectories and then managing rules from them. As Tab 4 shows, without online rule management, the generated manual can only slightly improve planning (from 88.1% to 90.7%). This is because more mundane mistakes and fewer direct successes will occur in the trajectories (the distributional shift problem), and the rules cannot be verified by feedback from the environment.

**Skill&Reflection Libraries:** Retrieving historical conclusions is essential for correct planning, as they record massive interacting details that can complement the rules. Without them, there will be

more errors in the details, and the success rate drops from 97.4% to 89.5%. However, as discussed previously, using plain experiences without inducing rules will lead to Path Dependency.

**Case-Conditional Prompts:** This strategy further improves the rule management process by reducing the hallucination, as evidenced by an increase in success rate from 93.8% to 97.4%. These prompts ensure that the Builder updates rules reasonably and grounded.

**Effect of Manual Formulation:** The final formulation of rules into a comprehensive manual contributed to the success rate of 97.4% and decreased average error steps, demonstrating the effectiveness of presenting rule-based knowledge in an organized and accessible format. It not only aids the Planner in mastering multiple rules but is also friendly for human reading.

# 5   Conclusion

In this paper, we introduce AutoManual, a framework significantly advancing LLM agents by enabling adaptability and continual learning through online rule optimization. Utilizing the structured rule system, AutoManual autonomously generates comprehensive manuals, achieving high success rates in benchmarks like ALFWorld and MiniWoB++. This approach reduces reliance on human-provided examples and expert interventions, illustrating a robust method for enhancing agent generalization and addressing the Path Dependency problem in diverse environments.

# 6   Acknowledgements

Thanks to Dr. Zhe Zeng for her invaluable assistance with the OpenAI API and GPT Plus services. This work was supported in part by The National Nature Science Foundation of China (Grant No: 62273303), in part by Yongjiang Talent Introduction Programme (Grant No: 2022A-240-G).

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

## Appendix

## A Limitations

Despite the significant contributions of the AutoManual framework, several limitations warrant further discussion. First, our method heavily relies on the capabilities of GPT-4-turbo to generate reliable rules, which may limit the framework's applicability with less advanced language models.

Secondly, the current implementation places all rules directly within the context of LLM, which, while effective in smaller or moderately complex environments, may not scale well to larger, more dynamic settings. For such expansive environments, integrating the rule system with Retrieval-Augmented Generation (RAG) techniques, similar to the approach taken by AutoGuide [2], could enhance performance by dynamically selecting relevant rules based on the context, thereby managing the cognitive load on the LLM more efficiently.

Thirdly, for complex and challenging tasks, the agents of AutoManual are insufficient in exploring the environment, as they only attempt solutions based on current knowledge. To enhance agents' exploration of unfamiliar environments, it may be necessary to endow the agent with curiosity [24] or combine it with tree search algorithms [32].

Lastly, there remains a challenge in ensuring that the Planner agent consistently adheres to the rules outlined in the manual. In practice, the Planner may sometimes ignore these rules (or have hallucinations about observations) and cannot generate the ideal solution code that can be applied to different situations. This indicates a need for additional mechanisms to enforce or verify rule compliance during operations. This issue underscores the potential necessity for developing more sophisticated methods to ensure rule adherence or to introduce more robust validation steps within the planning process.

## B  Broader Impacts

The AutoManual framework, leveraging LLM agents, presents positive and negative impacts on safety. On the positive side, by autonomously generating reliable manuals, our method enhances the predictability and reliability of LLM behaviors in dynamic environments, potentially reducing errors and increasing operational safety. However, relying on LLMs also introduces risks of unpredictable behaviors when agents encounter unanticipated scenarios or when rule adherence is not fully ensured.

Furthermore, the manuals generated by our method can be invaluable tools for human workers. They encapsulate a distilled form of interaction-based learning that can aid in training, provide decision support, and improve task efficiency in various domains. This can not only enhance productivity but also ensure that human workers are better informed and prepared to manage the complex systems with which they interact.

Lastly, AutoManual's ability to generate structured, context-aware manuals from interactive experiences suggests a promising avenue for constructing comprehensive knowledge bases for AI. These manuals can contribute to a global knowledge base shared by LLMs of different sizes, promoting broader AI research and development. It offers a method to systematically organize and retrieve complex interaction data in a way that is accessible and useful for both machines and humans.

## C  Difference with Prior Methods

We compare AutoManual with prior methods that extract rules from experiences, i.e., ExpeL [35] and AutoGuide [2], and discuss all differences here:

1) **Interactive Form.** ExpeL and AutoGuide, following Reflexion [16], each time output the thought and next-step action to interact with the environment. Our Planner agent interacts using free-form code. As evidenced by previous works [7, 25], using code as the plan is more efficient because the code will automatically perform the in-plan actions and require LLM responses far fewer times. Additionally, planning with code can enjoy the powerful programming capability of GPT. More importantly, compared with action sequences, code is easier to generalize to similar scenarios, facilitating the management.

2) **Online v.s. Offline.** ExpeL and AutoGuide extract rules from offline experiences of Reflexion agent. We update rules online by alternating rule practice and rule management. This online rule management can verify the reliability and applicability of the rules in a timely manner, forbidding rules to be armchair general. Additionally, online-update rules can help planners continuously improve their trajectories, making it easier for higher-quality success processes to emerge.

3) **Collaboration Between Agents.** Building rules online also enables collaboration between the Planner and Builder agents. In our AutoManual, the Planner is prompted to describe the rules considered and analyze special phenomena, which ease the management of the Builder. In contrast, the builder in ExpeL and AutoGuide can only receive action sequences from the Planner.

4) **Rule System.** In ExpeL and AutoGuide, each rule has only two attributes: the content and score. In our rule system, each rule has four attributes: "Type", "Content", "Example", and "Logs". These attributes provide a comprehensive representation of rules and facilitate the management and usage of rules. Moreover, we allow rules to build on top of other rules.

5) **Handling excessive rules.** ExpeL and AutoGuide utilize the rule scores to delete the low-score rules. However, we found that rule scores are unreliable in the experiments because the Builder tends to give overconfident scores to all rules. Instead, our AutoManual employs a Consolidator agent to merge and delete redundant rules.

## D  Implementation Details

In the building stage of all experiments, the maximum number of rules was set to 12 to balance the context length and rule diversity. We use OpenAI's Assistant API for all agents to save their history and prevent duplicate inputs. We set all LLMs' temperatures to 0 and maximum context length to 16000. Reflexion agents are allowed to try at most 3 trials for each task. For AdaPlanner and AutoManual, we allow the Planner agent to replan at most 3 times on ALFWorld and 6 times on

Table 5: The GPT models used for each method in our implementation when using GPT-3.5-turbo or GPT-4-turbo as test-time LLM. Here GPT-3.5-turbo denotes `gpt-3.5-turbo-1106` and GPT-4-turbo denotes `gpt-4-1106-preview`.

| Method | Test-Time LLM | |
| | **GPT-3.5-turbo** | **GPT-4-turbo** |
| --- | --- | --- |
| ReAct [33] | Actor: GPT-3.5-turbo | Actor: GPT-4-turbo |
| Reflexion [16] | Actor: GPT-3.5-turbo
Self-reflection: GPT-3.5-turbo | Actor: GPT-4-turbo
Self-reflection: GPT-4-turbo |
| ExpeL [35] | Offline Trajectory:
    Actor: GPT-3.5-turbo
    Self-reflection: GPT-3.5-turbo
Insight Extraction:
    Builder: GPT-4-turbo
Task Inference:
    Actor: GPT-3.5-turbo | Offline Trajectory:
    Actor: GPT-4-turbo
    Self-reflection: GPT-4-turbo
Insight Extraction:
    Builder: GPT-4-turbo
Task Inference:
    Actor: GPT-4-turbo |
| RCI [5] | Task/State/Agent grounding:
    GPT-3.5-turbo | Task/State/Agent grounding:
    GPT-4-turbo |
| Adaplanner [21] | Planner/Refiner: GPT-3.5-turbo
    `ask_LLM()`: GPT-3.5-turbo-instruct | Planner/Refiner: GPT-4-turbo
    `ask_LLM()`: GPT-3.5-turbo-instruct |
| AutoManual | Building Stage:
    Planner: GPT-4-turbo
    Builder: GPT-4-turbo
Formulator: GPT-4-turbo
Testing Stage:
    Planner: GPT-3.5-turbo | Building Stage:
    Planner: GPT-4-turbo
    Builder: GPT-4-turbo
Formulator: GPT-4-turbo
Testing Stage:
    Planner: GPT-4-turbo |

MiniWob++ in response to the environmental feedback. In the Building stage on ALFWorld, we use 6 tasks for each task type, a total of 36 tasks, by default. We shuffle all tasks, and when tasks in a task type succeed three times in a row during building, we consider that this task type has been solved and will no longer run it. The API call cost for building and formulating stages is about **$14** in total.

We made a slight modification to the text format of ALFWorld to make it more suitable for code: for each object, "object id" was changed to "object_id", and all preceding articles were removed. The maximum action step for a task is set to 50.

For the 9 tasks with feedback on Miniwob++, we find that "email-inbox-nl-turk" and "email-inbox-forward-nl-turk" are repetitions of "email-inbox" and "email-inbox-forward-nl". Therefore, we only used 7 task types in the building stage, while in the testing stage, all 9 types were evaluated. In the building stage, we run a total of 42 tasks (6 tasks for each type).

For all 53 tasks on Miniwob++, since running the building directly will lead to a large number of rules, and tasks without feedback have low variability, we adopt a two-step building strategy: we first run our building stage on 9 task types with feedback and then only update the skill and reflection libraries on 44 task types without feedback. The 4 examples for experiments on all task types are from the following task types: 'click-menu', 'enter-date', 'social-media-some'. We chose these tasks mainly because they are difficult for GPT-4-turbo to try out due to the lack of HTML feedback. For example, the HTML in 'enter-date' will not display the cursor but requires the cursor at the correct position to enter.

Environmental feedback on executing code in MiniWob++ is as follows: Whenever one of the actions is executed, we will log whether the action was executed successfully or failed. Finally, the results of these actions are concatenated together, and used as feedback of the code along with the HTML text.

For AdaPlanner [21] and RCI [5], we use their official code on GitHub to implement their methods. We fix their bugs on matching text patterns but still find their performance is much lower than they reported. This may be because they have not released the code used in their papers, or their methods are greatly affected by the GPT version (newer versions of GPT will be more creative).

# E   Presentation of Generated Manuals

We present the generated manual[3] on ALFWorld by AutoManual in Fig 5. As shown in the manual, the Formulator agent categorizes the rules into four categories: "Navigation and Search", "Object Interaction and Location Management", "Task-Specific Processes", and "Correctness and Validation" with clear introductions. We find our agents have found and recorded lots of important rules that directly impact task completion. For instance, *"the agent should include all possible locations in its search"* in rule_0, *"The agent can only hold one object at a time"* in rule_3, and *"When multiple items of the same type are present at a location, the agent may have to choose one to interact with or examine."* in rule_5. For tasks that can be solved by a fixed strategy, such as "Cool", "Heat", "Clean" and "Examine", AutoManual provides clear Success Process type rules. For the complex task, "Put Two", AutoManual conducts classified discussions in its Success Process rule.

**These demonstrate that AutoManual resolves the Path Dependency problem of skills by digging deeper into mechanisms, updating and incorporating success processes, and annotating important details.**

Moreover, we prompt the Builder to break down large phenomena into specific rules and derive deeper rules from them later. In the initial rules 9, we give a simple example of how to build rules upon rules. In the manual 5, we find that the Builder can correctly utilize rules recording basic mechanisms to build more complex rules. For example, rule_4 uses rule_3 to induce a solution: *"If all objects are found at the same location, handle them sequentially according to rule_3."*

However, we find the generated manual is still not perfect. There are some unnecessary duplicates between rules, such as rule_7 and rule_8, which are both Success Process type rules for "Examine" tasks, but they are divided into two rules.

Table 6: The success rate (%) of different methods on 9 task types with feedback of MiniWob++.

| Task Type | RCI [5] | AdaPlanner [21] | Planner+Lib. | AutoManual |
|---|---|---|---|---|
| Examples | 22 | 13 | 1 | 1 |
| *Test with GPT-3.5-turbo* | | | | |
| search-engine | 33.3 | 100.0 | 66.7 | 66.7 |
| tic-tac-toe | 22.2 | 27.8 | 16.7 | 33.3 |
| terminal | 55.6 | 100.0 | 100.0 | 100.0 |
| login-user-popup | 33.3 | 33.3 | 33.3 | 66.7 |
| guess-number | 11.1 | 22.2 | 11.1 | 94.4 |
| email-inbox | 77.8 | 88.9 | 83.3 | 100.0 |
| email-inbox-nl-turk | 61.1 | 94.4 | 77.8 | 100.0 |
| email-inbox-forward-nl | 61.1 | 83.3 | 94.4 | 100.0 |
| email-inbox-forward-nl-turk | 55.6 | 94.4 | 88.9 | 77.8 |
| Average | 45.6 | 71.6 | 63.6 | 82.2 |
| *Test with GPT-4-turbo* | | | | |
| search-engine | 44.4 | 100.0 | 100.0 | 100.0 |
| tic-tac-toe | 33.3 | 22.2 | 22.2 | 66.7 |
| terminal | 88.9 | 100.0 | 100.0 | 100.0 |
| login-user-popup | 38.9 | 33.3 | 72.2 | 100.0 |
| guess-number | 22.2 | 44.1 | 33.3 | 88.9 |
| email-inbox | 77.8 | 100.0 | 100.0 | 94.4 |
| email-inbox-nl-turk | 72.2 | 100.0 | 100.0 | 100.0 |
| email-inbox-forward-nl | 88.9 | 100.0 | 100.0 | 100.0 |
| email-inbox-forward-nl-turk | 77.8 | 66.7 | 94.4 | 100.0 |
| Average | 60.4 | 74.1 | 80.2 | 94.5 |

---

[3]The manual doesn't include Validation Logs of the rules, which are not visible for the Planner.

# Housekeeper Agent Interaction Manual

## Overview

This manual is intended to assist the housekeeper agent in the successful execution of tasks within a simulated environment. The rules provide guidance on navigating, searching the environment, interacting with objects, and managing task-specific processes, as well as ensuring the correctness of actions using code assertions.

## Navigation and Search

### Introduction

These rules provide guidance on how to search for objects, including the use of helper methods to streamline the process and ensure thoroughness.

### Included Rules

- **rule_0 (type="Special Mechanism")**: Objects can be found in unconventional locations, and the agent should **include all possible locations in its search**. For example, In epoch_9, the agent found a soapbar on the toilet, which is an unconventional location for storing such items.

- **rule_1 (type="Useful Helper Method")**: If there are multiple receptacles to be search, the agent can write and use 'find_object' method as shown in the example. For example,

```python
# Define helper method to find object that is needed
def find_object(agent, recep_to_check, object_name):
    for receptacle in recep_to_check:
        observation = agent.go_to(receptacle)
        # Check if we need to open the receptacle. If we do, open it.
        if 'closed' in observation:
            observation = agent.open(receptacle)
        # Check if the object is in/on the receptacle.
        if object_name in observation:
            object_ids = get_object_with_id(observation, object_name)
            return object_ids, receptacle
    return None, None

# Use assertions to validate each step
assert object_ids is not None, "Error: Could not find the object."
```

## Object Interaction and Location Management

### Introduction

These rules inform the agent on how to interact with objects, from taking and placing items to handling multiple items of the same type. Proper location management is crucial for successful task execution.

### Included Rules

- **rule_2 (type="Special Phenomena")**: When using a microwave, the agent can interact with it (e.g., heat an object) even if there is another object inside, the agent is holding something, and the microwave door is not explicitly mentioned to be open.

  For example, In epoch_1, the agent was able to heat the mug with the microwave even though there was an egg inside the microwave and the agent was holding the mug.

- **rule_3 (type="Special Mechanism")**: The agent can only hold one object at a time and must put down any held object before taking another.

  For example, In epoch_2, the agent was holding statue_4 and attempted to take statue_3 without putting down statue_4 first, resulting in a 'Nothing happens' observation.

- **rule_4 (type="Success Process")**: When tasked with placing multiple objects in/on a receptacle, the agent can either collect all objects before attempting to place them or find and place them one by one, ensuring they revisit locations with multiple objects if necessary. If all objects are found at the same location, handle them sequentially according to rule_3.

  For example, In epoch_15, the agent should have revisited sidetable_1 to collect the second pencil before attempting to place it in coffeetable_1. In epoch_23, the agent failed to collect all required statues from coffeetable_1 because it did not revisit, is also addressed by this rule.

Figure 5: The Generated Manual for ALFWorld: Part 1.

- **rule_5 (type="Special Mechanism")**: The agent must interact with a receptacle to observe its contents, which includes going to the receptacle and opening it if it is closed. Before performing a put or take action, the agent must ensure it is at the correct location. When multiple items of the same type are present at a location, the agent may have to choose one to interact with or examine.

  For example, In epoch_16, the agent had to open several closed cabinets (e.g., cabinet_1, cabinet_2) to find items such as the mug. In epoch_21, the agent observed multiple alarm clocks on desk_1 and selected one ('alarmclock_4') to interact with.

## Task-Specific Processes

### Introduction

This category outlines the steps required to complete specific tasks, such as heating, cooling, and examining objects with another object's assistance.

### Included Rules

- **rule_6 (type="Success Process")**: If the task involves cooling or heating an object before placing it, the steps are: (1) search for the object using 'find_object' in rule_1, (2) take the object, (3) cool/heat it as required, (4) go to the target receptacle, and (5) put the object. Ensure the agent's location and the state of the environment are updated after each action. For example,

```
# For example, to cool a mug and put it in a coffeemachine:
# [Step 1] Use 'find_object' method to search all receptacles
# [Step 2] Take the mug
# [Step 3] Go to the fridge, open it if necessary, and cool the mug
# [Step 4] Go to the coffeemachine and put the cooled mug in it
```

- **rule_7 (type="Success Process")**: When tasked with examining an object under a desklamp, the agent should first find the desklamp and the object, ensure the desklamp is on, take the object, and then use the desklamp to examine the object. For example,

```
# [Step 1] Use 'find_object' method to search for the desklamp and the object
# [Step 2] Make sure the desklamp is on
# [Step 3] Take the object
# [Step 4] Use the desklamp to examine the object.
```

- **rule_8 (type="Success Process")**: When tasked to look at an object under a desklamp, ensure the lamp is on before using it to examine the object. For example,

```
# [Step 4] Go to the desklamp's location and turn it on if it's not already on
observation = agent.go_to(receptacle_with_desklamp)
observation = agent.use(found_desklamp)
assert 'turn on' in observation or 'already on' in observation, 'Error in [Step 4]: Failed to use the desklamp.'
# [Step 5] Similarly, search for the alarm clock and take it.
# [Step 6] With the desklamp on, examine the alarm clock using the desklamp.
```

## Correctness and Validation

### Introduction

Instructions on asserting code to confirm state changes and enhance the reliability of the agent's actions.

### Included Rules

- **rule_9 (type="Corrected Error")**: Assertions in the agent's code should confirm state changes such as location or held objects, rather than rely on specific phrases in observations.

  For example, Instead of asserting 'You are at' in the observation, the agent should assert the location and held object state changes. Also, when handling multiple required objects at the same location, the agent should manage them sequentially without unnecessary variables.

Figure 6: The Generated Manual for ALFWorld: Part 2.

Table 7: Ablation study of Rule System on ALFWorld.

| Method | Success Rate (%) Testing with | |
|---|---|---|
| | **GPT-3.5-turbo** | **GPT-4-turbo** |
| AutoManual | 86.2 | 97.4 |
| AutoManual without "Type" | 74.6 | 91.5 |
| AutoManual without "Example" | 76.8 | 92.7 |
| AutoManual without "Validation Logs" | 85.4 | 97.0 |
| AutoManual without "Useful Helper Method" | 79.6 | 94.3 |
| AutoManual without Cooperation between Agents | 78.8 | 93.8 |
| *Case-Conditioned Prompts* without Classification | 83.2 | 95.6 |

Table 8: Sensitivity Analysis of Examples and Initial Rules on 9 task types with feedback of MiniWob++. AutoManual uses a human example of `search-engine` or `enter-text` task.

| Method | Success Rate (%) Testing with | |
|---|---|---|
| | **GPT-3.5-turbo** | **GPT-4-turbo** |
| AutoManual | 82.2 | 94.5 |
| AutoManual with `enter-text` Example | 78.3 | 92.8 |

# F  More Experiments

## F.1  Ablation Study of Rule System

The well-structured rule system is essential for the rule management and usage. We perform the ablation study of our rule system in Tab 7.

- *AutoManual without "Type"*: we remove the "Type" attribute of each rule in the rule system and the instruction for the Builder to manage various types of rules. As shown in Tab 7, without the "Type" attribute, the performance drops significantly, from 86.2% to 74.6% and 97.4% to 91.5%, as the Builder cannot manage specific types of rules and loses specific instruction on each rule.

- *AutoManual without "Example"*: we remove the "Example" attribute of each rule in the rule system. The performance also drops by a large margin, as the Builder cannot specify necessary details in rules, and rules with lower understandability can sometimes be misleading to the Planner.

- *AutoManual without "Validation Logs"* has little impact on performance, but "Validation Logs" of rules are very useful during debugging.

We also conducted some detailed ablation studies of our AutoManual in Tab 7.

- Helper methods defined by humans and the Planner can serve as the solution for a subgoal of a task that can be used in multiple tasks. *AutoManual without "Useful Helper Method"*: we remove the "Useful Helper Method" in the "Type" attribute, the human example, and initial rules, and we no longer encourage the Planner to write helper functions when writing code. The performance drops from 86.2% to 79.6% and 97.4% to 94.3%, demonstrating that extracting reusable helper methods can facilitate the programming.

- *AutoManual without Cooperation between Agents*: we remove the Planner's thoughts in the trajectory when providing it to the Builder and no longer require the Planner to output Relevant Rules. The performance decreases significantly, indicating the thoughts and conclusions of the Planner are helpful for the Builder to manage rules.

- *Case-Conditioned Prompts* without Classification: we remove the requirement to classify the error reason to "Imperfect Rules" or "Imperfect Agents" and only use the prompts for "Imperfect Rules" during *case-conditioned prompting*. As demonstrated in Tab 7, the results are inferior to using all 5 cases because the analysis and classification of error reason can boost the Builder to manage rules accurately.

## F.2 Sensitivity Analysis of Examples and Initial Rules

We analyze the sensitivity of our rule optimization to the initial condition, i.e., the human example and initial rules. In the experiments on 9 task types with feedback of MiniWob++, we use an example of `search-engine` task and the corresponding initial rules, which are shown in Listing 8 and 9. In these initial rules, we extract the successful process of `search-engine` task and a helper method to turn to the next page. Here, we provide AutoManual with an example of `enter-text` task, the simplest task type in MiniWob++, as shown in Listing 10.

The results in Tab 8 demonstrate that even if given less initial knowledge, AutoManual can still automatically learn the required knowledge through online optimization, which shows that AutoManual has good robustness to the initial rules. In addition, we found that the knowledge it learned about `search-engine` is highly similar to the standard knowledge provided by humans, such as the need to turn pages for the search index larger than 3.

## G   Case Study

### G.1   An Example of Path Dependency Problem

In the following example, the existing skill can correct solve the "Put two" task when the two objects occur at the same receptacles. However, in the new scenario, the Planner still replicates the strategy but fails to complete the task, as the target objects are at different receptacles.

Listing 1: An Example of Path Dependency Problem of the Planner on ALFWorld

```
Here is the code for a relevant skill:
You are in the middle of a room. Looking quickly around you, you see bed_1, desk_1, drawer_5, drawer_4,
 drawer_3, drawer_2, drawer_1, dresser_1, and garbagecan_1.
The task is to: put two cellphone in bed.
```python
# Helper method to manually parse the observation for cellphones
def parse_for_cellphones(observation):
    # This is a simple parser that looks for the word 'cellphone' followed by an underscore and a number
    return [word for word in observation.split() if 'cellphone' in word]

# Define the list of receptacles to search for the cellphones.
receptacles_to_check = [
    'drawer_1', 'drawer_2', 'drawer_3', 'drawer_4', 'drawer_5', 'dresser_1'
]

# Initialize a list to keep track of found cellphones.
found_cellphones = []

# Search each receptacle for the cellphones.
for receptacle in receptacles_to_check:
    observation = agent.go_to(receptacle)
    # If the receptacle is closed, open it.
    if 'closed' in observation:
        observation = agent.open(receptacle)
    # Manually parse the observation for cellphones.
    found_cellphones.extend(parse_for_cellphones(observation))
    # If two cellphones are found, stop searching.
    if len(found_cellphones) >= 2:
        break

# Ensure two cellphones were found.
assert len(found_cellphones) >= 2, 'Error: Less than 2 cellphones found in the environment.'

# Take and place the cellphones in the bed.
for cellphone in found_cellphones[:2]:  # Take only two cellphones
    # Go to the receptacle where the cellphone was found.
    observation = agent.go_to(receptacle)
    # Take the cellphone.
    observation = agent.take_from(cellphone, receptacle)
    # Go to the bed and put the cellphone in it.
    observation = agent.go_to('bed_1')
    observation = agent.put_in_or_on(cellphone, 'bed_1')
```
Please pay close attention to the process and details of this successful code when writing code. Also, be
 aware of potential randomness (the current environment may differ from this one).

Initial observation and the task:
```

You are in the middle of a room. Looking quickly around you, you see armchair_1, cabinet_4, cabinet_3,
  cabinet_2, cabinet_1, drawer_5, drawer_4, drawer_3, drawer_2, drawer_1, dresser_1, garbagecan_1, safe_1,
  shelf_12, shelf_11, shelf_10, shelf_9, shelf_8, shelf_7, shelf_6, shelf_5, shelf_4, shelf_3, shelf_2,
  shelf_1, sidetable_1, and sofa_1.
Your task is to: put two pillow in sofa.

### Understanding of the task:
The task is to find two pillows in the environment and place them in or on the sofa. The environment
  contains various receptacles and surfaces where the pillows might be located.

### Plan for the task:
1. Define a list of all receptacles and surfaces to check for pillows.
2. Search each receptacle for pillows.
3. Take two pillows once found.
4. Move to the sofa's location.
5. Place each pillow on the sofa.

Here is the code for the task:

```python
# Helper method to manually parse the observation for pillows
def parse_for_pillows(observation):
    # This is a simple parser that looks for the word 'pillow' followed by an underscore and a number
    return [word for word in observation.split() if 'pillow' in word]

# Define the list of receptacles to search for the pillows.
receptacles_to_check = [
    'cabinet_1', 'cabinet_2', 'cabinet_3', 'cabinet_4', 'drawer_1', 'drawer_2',
    'drawer_3', 'drawer_4', 'drawer_5', 'dresser_1', 'shelf_1', 'shelf_2', 'shelf_3',
    'shelf_4', 'shelf_5', 'shelf_6', 'shelf_7', 'shelf_8', 'shelf_9', 'shelf_10',
    'shelf_11', 'shelf_12', 'sidetable_1'
]

# Initialize a list to keep track of found pillows.
found_pillows = []

# Search each receptacle for the pillows.
for receptacle in receptacles_to_check:
    observation = agent.go_to(receptacle)
    # If the receptacle is closed, open it.
    if 'closed' in observation:
        observation = agent.open(receptacle)
    # Manually parse the observation for pillows.
    found_pillows.extend(parse_for_pillows(observation))
    # If two pillows are found, stop searching.
    if len(found_pillows) >= 2:
        break

# Ensure two pillows were found.
assert len(found_pillows) >= 2, 'Error: Less than 2 pillows found in the environment.'

# Take and place the pillows in the sofa.
for pillow in found_pillows[:2]:  # Take only two pillows
    # Go to the receptacle where the pillow was found.
    observation = agent.go_to(receptacle)
    # Take the pillow.
    observation = agent.take_from(pillow, receptacle)
    # Go to the sofa and put the pillow in it.
    observation = agent.go_to('sofa_1')
    observation = agent.put_in_or_on(pillow, 'sofa_1')
```

obs_1: Act: agent.go_to('cabinet_1'). Obs: cabinet_1 is closed. You are at cabinet_1 and holding nothing.
obs_2: Act: agent.open('cabinet_1'). Obs: You open cabinet_1. In cabinet_1, you see tissuebox_1. You are
  at cabinet_1 and holding nothing.
obs_3: Act: agent.go_to('cabinet_2'). Obs: Nothing happens. You are at cabinet_1 and holding nothing.
obs_4: Act: agent.go_to('cabinet_3'). Obs: cabinet_3 is closed. You are at cabinet_3 and holding nothing.
obs_5: Act: agent.open('cabinet_3'). Obs: You open cabinet_3. In cabinet_3, you see box_3, and tissuebox_2
  . You are at cabinet_3 and holding nothing.
obs_6: Act: agent.go_to('cabinet_4'). Obs: Nothing happens. You are at cabinet_3 and holding nothing.
obs_7: Act: agent.go_to('drawer_1'). Obs: drawer_1 is closed. You are at drawer_1 and holding nothing.
obs_8: Act: agent.open('drawer_1'). Obs: You open drawer_1. In drawer_1, you see nothing. You are at
  drawer_1 and holding nothing.
obs_9: Act: agent.go_to('drawer_2'). Obs: drawer_2 is closed. You are at drawer_2 and holding nothing.
obs_10: Act: agent.open('drawer_2'). Obs: You open drawer_2. In drawer_2, you see nothing. You are at
  drawer_2 and holding nothing.
obs_11: Act: agent.go_to('drawer_3'). Obs: drawer_3 is closed. You are at drawer_3 and holding nothing.
obs_12: Act: agent.open('drawer_3'). Obs: You open drawer_3. In drawer_3, you see nothing. You are at
  drawer_3 and holding nothing.
obs_13: Act: agent.go_to('drawer_4'). Obs: drawer_4 is closed. You are at drawer_4 and holding nothing.

```
obs_14: Act: agent.open('drawer_4'). Obs: You open drawer_4. In drawer_4, you see nothing. You are at
  drawer_4 and holding nothing.
obs_15: Act: agent.go_to('drawer_5'). Obs: drawer_5 is closed. You are at drawer_5 and holding nothing.
obs_16: Act: agent.open('drawer_5'). Obs: You open drawer_5. In drawer_5, you see nothing. You are at
  drawer_5 and holding nothing.
obs_17: Act: agent.go_to('dresser_1'). Obs: On dresser_1, you see box_2, laptop_1, statue_1, watch_2, and
  watch_1. You are at dresser_1 and holding nothing.
obs_18: Act: agent.go_to('shelf_1'). Obs: On shelf_1, you see nothing. You are at shelf_1 and holding
  nothing.
obs_19: Act: agent.go_to('shelf_2'). Obs: On shelf_2, you see nothing. You are at shelf_2 and holding
  nothing.
obs_20: Act: agent.go_to('shelf_3'). Obs: Nothing happens. You are at shelf_2 and holding nothing.
obs_21: Act: agent.go_to('shelf_4'). Obs: On shelf_4, you see nothing. You are at shelf_4 and holding
  nothing.
obs_22: Act: agent.go_to('shelf_5'). Obs: On shelf_5, you see newspaper_2. You are at shelf_5 and holding
  nothing.
obs_23: Act: agent.go_to('shelf_6'). Obs: On shelf_6, you see nothing. You are at shelf_6 and holding
  nothing.
obs_24: Act: agent.go_to('shelf_7'). Obs: On shelf_7, you see watch_3. You are at shelf_7 and holding
  nothing.
obs_25: Act: agent.go_to('shelf_8'). Obs: On shelf_8, you see nothing. You are at shelf_8 and holding
  nothing.
obs_26: Act: agent.go_to('shelf_9'). Obs: On shelf_9, you see nothing. You are at shelf_9 and holding
  nothing.
obs_27: Act: agent.go_to('shelf_10'). Obs: On shelf_10, you see nothing. You are at shelf_10 and holding
  nothing.
obs_28: Act: agent.go_to('shelf_11'). Obs: On shelf_11, you see nothing. You are at shelf_11 and holding
  nothing.
obs_29: Act: agent.go_to('shelf_12'). Obs: On shelf_12, you see cellphone_1. You are at shelf_12 and
  holding nothing.
obs_30: Act: agent.go_to('sidetable_1'). Obs: On sidetable_1, you see creditcard_1, keychain_1, and
  newspaper_1. You are at sidetable_1 and holding nothing.
Execution error:
Error: Less than 2 pillows found in the environment.
Current state: You are at sidetable_1 and holding nothing.
```

## G.2 A Trajectory of The Building Stage

Here, we present a trajectory of the Planner and the corresponding responses of the Builder for a task in ALFWorld. In this "Indirect Success" trajectory, the Planner made the mistake of heating the mug without having it next to the microwave. The Planner then corrected the mistake and provided organized code in its conclusion. After observing this trajectory, the Builder keenly found the cause of the error: the existing "Success Process" rule did not indicate that the agent must go to the object before interacting with the object. The Builder updated the "Success Process" rule and added a "Corrected Error" rule to clarify the mistake.

Listing 2: A Trajectory of the Planner and Builder on ALFWorld

```
Initial observation and the task:
You are in the middle of a room. Looking quickly around you, you see cabinet_13, cabinet_12, cabinet_11,
  cabinet_10, cabinet_9, cabinet_8, cabinet_7, cabinet_6, cabinet_5, cabinet_4, cabinet_3, cabinet_2,
  cabinet_1, coffeemachine_1, countertop_1, diningtable_1, drawer_1, fridge_1, garbagecan_1, microwave_1,
  shelf_3, shelf_2, shelf_1, sinkbasin_1, stoveburner_4, stoveburner_3, stoveburner_2, stoveburner_1, and
  toaster_1.
Your task is to: heat some mug and put it in cabinet.
Planner:
### Understanding of the observation:
The agent is in a room with numerous cabinets, appliances, and surfaces. The task is to heat a mug and
  then place it in a cabinet. This requires finding a mug, heating it, and then placing it in any cabinet.

### Rules to consider:
- **rule_0**: Ensure all receptacles and surfaces are considered when searching for the mug.
- **rule_1**: Use the 'find_object' method to search for the mug.
- **rule_2**: After finding the mug, heat it and then put it in a cabinet.
- **rule_3**: Open closed receptacles to search for objects.

### Plan for the task:
1. Use the 'find_object' method to search for a mug in all receptacles and surfaces.
2. Once the mug is found, take it.
3. Heat the mug with an appropriate appliance (e.g., microwave).
4. Find an empty cabinet or any cabinet to place the heated mug in.
5. Place the mug in the cabinet.

Here is the code for the task:
```python
```

```
# [Step 1] Define the helper method 'find_object' as provided in rule_1
def find_object(agent, recep_to_check, object_name):
    for receptacle in recep_to_check:
        observation = agent.go_to(receptacle)
        # Check if we need to open the receptacle. If we do, open it.
        if 'closed' in observation:
            observation = agent.open(receptacle)
        # Check if the object is in/on the receptacle.
        if object_name in observation:
            object_ids = get_object_with_id(observation, object_name)
            return object_ids, receptacle
    return None, None

# [Step 2] Get a sorted list of all receptacles and surfaces to check for a mug
recep_to_check = ['cabinet_1', 'cabinet_2', 'cabinet_3', 'cabinet_4', 'cabinet_5', 'cabinet_6', 'cabinet_7
', 'cabinet_8', 'cabinet_9', 'cabinet_10', 'cabinet_11', 'cabinet_12', 'cabinet_13', 'countertop_1', '
 diningtable_1', 'drawer_1', 'fridge_1', 'garbagecan_1', 'microwave_1', 'shelf_1', 'shelf_2', 'shelf_3', '
 sinkbasin_1', 'stoveburner_1', 'stoveburner_2', 'stoveburner_3', 'stoveburner_4', 'toaster_1', '
 coffeemachine_1']
object_ids, receptacle_with_mug = find_object(agent, recep_to_check, 'mug')
# Assert that we have found a mug
assert object_ids is not None, f'Error in [Step 2]: There is no mug in/on {recep_to_check}.'

# [Step 3] Take the mug
found_mug = object_ids[0]
observation = agent.take_from(found_mug, receptacle_with_mug)
# Assert that the agent is holding the mug
assert agent.holding == found_mug, f'Error in [Step 3]: I cannot take {found_mug} from {
 receptacle_with_mug}.'

# [Step 4] Heat the mug with an appropriate appliance (e.g., microwave)
observation = agent.heat_with(found_mug, 'microwave_1')
# Assert that the mug has been heated
assert 'heat' in observation and found_mug in observation, f'Error in [Step 4]: I cannot heat {found_mug}
 with microwave_1.'

# [Step 5] Find an empty cabinet or any cabinet to place the heated mug in
# Since the task does not specify which cabinet to use, we can choose the first one
observation = agent.go_to('cabinet_1')
# Check if the cabinet is closed. If so, open it.
if 'closed' in observation:
    observation = agent.open('cabinet_1')
observation = agent.put_in_or_on(found_mug, 'cabinet_1')
# Assert that the mug is put in the cabinet
assert 'put' in observation and found_mug in observation, f'Error in [Step 5]: I cannot put {found_mug} in
  cabinet_1.'
'''
```

Feedback including errors:

```
obs_1: Act: agent.go_to('cabinet_1'). Obs: cabinet_1 is closed. You are at cabinet_1 and holding nothing.
obs_2: Act: agent.open('cabinet_1'). Obs: You open cabinet_1. In cabinet_1, you see nothing. You are at
 cabinet_1 and holding nothing.
...
obs_33: Act: agent.go_to('shelf_1'). Obs: On shelf_1, you see mug_1. You are at shelf_1 and holding
 nothing.
obs_34: Act: agent.take_from('mug_1', 'shelf_1'). Obs: You take mug_1 from shelf_1. You are at shelf_1 and
  holding mug_1.
obs_35: Act: agent.heat_with('mug_1', 'microwave_1'). Obs: Nothing happens. You are at shelf_1 and holding
  mug_1.
Execution error:
```

Error in [Step 4]:  I cannot heat mug_1 with microwave_1.

Current state: You are at shelf_1 and holding mug_1.

Planner's replanning:

### Understanding of the current observations:
The agent successfully found a mug on shelf_1 and took it. However, when attempting to heat the mug with
 the microwave, the action failed with "Nothing happens." This indicates that the agent tried to use the
 microwave while not being at the microwave's location. The agent must go to the microwave before
 attempting to heat the mug.

### Mistakes in the previous code and the reason for errors:
The error occurred because the agent attempted to heat the mug without being at the microwave's location.
 The precondition for heating an object with a receptacle is that the agent must be at the same location
 as the receptacle.

### Rules to consider:
- **rule_0**: Ensure all receptacles and surfaces are considered when searching for the mug.
- **rule_1**: Use the 'find_object' method to search for the mug.
- **rule_2**: After finding the mug, heat it and then put it in a cabinet.
- **rule_3**: Open closed receptacles to search for objects.

```
### Plan for the task:
1. Since the mug has already been found and is being held by the agent, proceed to the microwave.
2. Open the microwave if it is closed.
3. Heat the mug with the microwave.
4. Find an empty cabinet or any cabinet to place the heated mug in.
5. Place the mug in the cabinet.

Here is the updated code for the task:
'''python
# [Step 4] Go to the microwave and open it if it's closed
observation = agent.go_to('microwave_1')
if 'closed' in observation:
    observation = agent.open('microwave_1')
# Assert that the agent is at the microwave and it's open
assert 'microwave_1' in observation and 'closed' not in observation, f'Error in [Step 4]: Microwave is not
  open or not at microwave_1.'

# [Step 5] Heat the mug with the microwave
observation = agent.heat_with('mug_1', 'microwave_1')
# Assert that the mug has been heated
assert 'heat' in observation and 'mug_1' in observation, f'Error in [Step 5]: I cannot heat mug_1 with
  microwave_1.'

# [Step 6] Find an empty cabinet or any cabinet to place the heated mug in
# Since the task does not specify which cabinet to use, we can choose the first one that is empty or
  available
# We already know that cabinets 1 through 10 are empty or contain other items, so we can choose cabinet_1
  for simplicity
observation = agent.go_to('cabinet_1')
if 'closed' in observation:
    observation = agent.open('cabinet_1')
# Assert that the agent is at cabinet_1 and it's open
assert 'cabinet_1' in observation and 'closed' not in observation, f'Error in [Step 6]: Cabinet_1 is not
  open or not at cabinet_1.'

# [Step 7] Place the mug in the cabinet
observation = agent.put_in_or_on('mug_1', 'cabinet_1')
# Assert that the mug is put in the cabinet
assert 'put' in observation and 'mug_1' in observation, f'Error in [Step 7]: I cannot put mug_1 in
  cabinet_1.'
'''
```

```
obs_36: Act: agent.go_to('microwave_1'). Obs: In microwave_1, you see egg_2. You are at microwave_1 and
  holding mug_1.
obs_37: Act: agent.heat_with('mug_1', 'microwave_1'). Obs: You heat mug_1 using microwave_1. You are at
  microwave_1 and holding mug_1.
obs_38: Act: agent.go_to('cabinet_1'). Obs: In cabinet_1, you see nothing. You are at cabinet_1 and
  holding mug_1.
obs_39: Act: agent.put_in_or_on('mug_1', 'cabinet_1'). Obs: You put mug_1 in/on cabinet_1. You are at
  cabinet_1 and holding nothing. This epoch is done. Succeed: True
Current state: You are at cabinet_1 and holding nothing.
```

Planner's conclusion:

```
### Summary of misunderstandings and mistakes:
Initially, I attempted to heat the mug without being at the microwave's location, which is why the action
  failed with "Nothing happens." The correct procedure is to first go to the microwave before attempting to
   heat any object. After correcting this mistake, the agent successfully heated the mug and placed it in
   cabinet_1.

### Organized code block:
'''python
# [Helper Method] Define the helper method 'find_object' as provided in rule_1
def find_object(agent, recep_to_check, object_name):
    for receptacle in recep_to_check:
        observation = agent.go_to(receptacle)
        # Check if we need to open the receptacle. If we do, open it.
        if 'closed' in observation:
            observation = agent.open(receptacle)
        # Check if the object is in/on the receptacle.
        if object_name in observation:
            object_ids = get_object_with_id(observation, object_name)
            return object_ids, receptacle
    return None, None

# [Step 1] Get a sorted list of all receptacles and surfaces to check for a mug
recep_to_check = ['cabinet_1', 'cabinet_2', 'cabinet_3', 'cabinet_4', 'cabinet_5', 'cabinet_6', 'cabinet_7
  ', 'cabinet_8', 'cabinet_9', 'cabinet_10', 'cabinet_11', 'cabinet_12', 'cabinet_13', 'countertop_1', '
  diningtable_1', 'drawer_1', 'fridge_1', 'garbagecan_1', 'microwave_1', 'shelf_1', 'shelf_2', 'shelf_3', '
  sinkbasin_1', 'stoveburner_1', 'stoveburner_2', 'stoveburner_3', 'stoveburner_4', 'toaster_1', '
  coffeemachine_1']
object_ids, receptacle_with_mug = find_object(agent, recep_to_check, 'mug')
```

```
# Assert that we have found a mug
assert object_ids is not None, f'Error in [Step 1]: There is no mug in/on {recep_to_check}.'

# [Step 2] Take the mug
found_mug = object_ids[0]
observation = agent.take_from(found_mug, receptacle_with_mug)
# Assert that the agent is holding the mug
assert agent.holding == found_mug, f'Error in [Step 2]: I cannot take {found_mug} from {
 receptacle_with_mug}.'

# [Step 3] Go to the microwave and open it if it's closed
observation = agent.go_to('microwave_1')
if 'closed' in observation:
    observation = agent.open('microwave_1')
# Assert that the agent is at the microwave and it's open
assert 'microwave_1' in observation and 'closed' not in observation, f'Error in [Step 3]: Microwave is not
   open or not at microwave_1.'

# [Step 4] Heat the mug with the microwave
observation = agent.heat_with(found_mug, 'microwave_1')
# Assert that the mug has been heated
assert 'heat' in observation and found_mug in observation, f'Error in [Step 4]: I cannot heat {found_mug}
 with microwave_1.'

# [Step 5] Find an empty cabinet or any cabinet to place the heated mug in
# Since the task does not specify which cabinet to use, we can choose the first one that is empty or
 available
# We already know that cabinets 1 through 10 are empty or contain other items, so we can choose cabinet_1
 for simplicity
observation = agent.go_to('cabinet_1')
if 'closed' in observation:
    observation = agent.open('cabinet_1')
# Assert that the agent is at cabinet_1 and it's open
assert 'cabinet_1' in observation and 'closed' not in observation, f'Error in [Step 5]: Cabinet_1 is not
 open or not at cabinet_1.'

# [Step 6] Place the mug in the cabinet
observation = agent.put_in_or_on(found_mug, 'cabinet_1')
# Assert that the mug is put in the cabinet
assert 'put' in observation and found_mug in observation, f'Error in [Step 6]: I cannot put {found_mug} in
   cabinet_1.'
'''
```

`Builder's classification:`

Analyzing the scenario, we can identify the root cause of the observed mistakes and relate them to the
 existing rules:

1. The agent initially attempted to heat the mug without being at the microwave's location, which resulted
   in the action "Nothing happens." This mistake was due to the agent not following the precondition for
  the 'heat_with' action, which requires the agent to be at the same location as the receptacle (in this
  case, the microwave).

2. The existing rules related to the mistakes are:
    - **rule_1**: This rule provides a 'find_object' method to search for objects, which the agent used
      correctly to find the mug.
    - **rule_2**: This 'Success Process' rule outlines the steps to put an object in a receptacle after
      finding and manipulating it. However, it does not explicitly state that the agent must be at the
      location of the appliance to use it, which is relevant for the heating step.
    - **rule_3**: This rule reminds the agent to open closed receptacles to search for objects, which the
      agent did correctly.

3. There is a 'Success Process' rule applicable to this type of task (rule_2), but it does not explicitly
 cover the precondition for heating an object, which is to be at the same location as the appliance.

4. `The fault in this scenario can be attributed to:`

   `*Imperfect Rules*:  The existing 'Success Process' rule (rule_2) does not fully document the`
   `precondition for using appliances, such as the microwave.  It does not state that the agent must be at`
   `the location of the appliance to use it for heating an object.`

Based on the analysis, the conclusion is that the fault stems from *Imperfect Rules*. The rules need to be
  updated to include the precondition for using appliances, which is a critical step in the 'Success
  Process' for tasks involving heating objects.

`Builder:`

Potential Rules:
1. *Corrected Error*: When attempting to use an appliance (e.g., microwave), the agent must be at the same
   location as the appliance. This rule is related to the existing 'Success Process' rule (rule_2), but it
  adds a specific precondition for using appliances that was not previously documented.

```
2. *Success Process*: The existing 'Success Process' rule (rule_2) should be updated to include the
   precondition of being at the location of the appliance before using it. This is a critical step in tasks
   involving heating objects.

Check Difference:
- The potential 'Corrected Error' rule targets the specific precondition for using appliances, which is a
  new addition and not covered by existing rules.
- The update to the 'Success Process' rule expands on the existing steps to include the precondition for
  using appliances, which is a refinement of the existing rule.

Check Existing Rules:
- Rule_2 needs to be updated to include the precondition for using appliances, as this was the key
  oversight that led to the agent's initial error.

Codes:
```python
# Update the existing 'Success Process' rule to include the precondition for using appliances
rule_system.update_rule(
    rule_id='rule_2',
    rule="If the task is to heat an object and put it in a receptacle, the agent must first find the
      object, ensure they are at the location of the heating appliance before using it, and then place the
      object in the target receptacle.",
    example="""# For example, to heat a mug and put it in a cabinet:
# [Step 1] Use 'find_object' method to search all receptacles
# [Step 2] Take the mug
# [Step 3] Go to the heating appliance (e.g., microwave) and ensure you are at its location
# [Step 4] Heat the mug with the appliance
# [Step 5] Find an empty cabinet or any cabinet to place the heated mug in
# [Step 6] Place the mug in the cabinet""",
    validation_record="Updated to include the precondition for using appliances based on the error and
      subsequent success in epoch_1."
)

# Write a new 'Corrected Error' rule for the precondition of using appliances
rule_system.write_rule(
    rule="Before using an appliance (e.g., microwave), the agent must go to the location of the appliance
      .",
    type="Corrected Error",
    example="""# Corrective code:
# Go to the microwave before attempting to heat the mug
observation = agent.go_to('microwave_1')
# Heat the mug with the microwave
observation = agent.heat_with('mug_1', 'microwave_1')""",
    validation_record="Derived from the error and correction in epoch_1 where the agent initially failed
      to heat the mug due to not being at the microwave's location."
)

rule_system.stop_generating()
```
```

# H  Prompts and Examples

## H.1  Prompts for Planner Agent

Listing 3: System Prompts for Planner Agent

```
[Role]
You are an excellent household agent in a simulated environment. Given a household
 environment and a task, you write Python code to accomplish the task.

[Actions]
For each task, an agent is created in an environment, and the initial observation
 and global memory are printed.

# The agent is an instance of Agent class, which includes the state of the agent (
 its location, what it's holding) and the actions it can take.

class Agent:
    def __init__(self, env: InteractiveEnvEngine):
        self.location = env.agent_location
        self.holding = "nothing"
        ...
```

```python
    # Here are the admissible actions the agent can take; all action functions
     return an observation string of the result of the action. If the precondition
     of the action is not met, its observation will include "Nothing happens".

    # Go to a receptacle and update the agent's location.
    # For example, 'On countertop_1, you see candle_1, cloth_2, and soapbar_1.' =
     go_to('countertop_1')
    # For example, 'On sidetable_2, you see nothing.' = go_to('sidetable_2')
    def go_to(self, receptacle):
        ...

    # Open a receptacle and observe its contents.
    # For example, 'You open cabinet_1. In cabinet_1, you see cloth_1.' = open('
     cabinet_1')
    def open(self, receptacle):
        ...

    # Close an opened receptacle.
    # For example, 'You close cabinet_1.' = close('cabinet_1')
    def close(self, receptacle):
        ...

    # Take an object from a receptacle if the agent is not holding anything.
    # For example, 'You take soapbar_1 from towelholder_1.' = take_from('soapbar_1',
      'towelholder_1')
    def take_from(self, object, receptacle):
        ...

    # Put an object in or on a receptacle if the agent is holding it.
    # For example, 'You put soapbar_1 in/on cabinet_1.' = put_in_or_on('soapbar_1',
     'cabinet_1')
    def put_in_or_on(self, object, receptacle):
        ...

    # Use a lamp.
    # For example, 'You turn on desklamp_1.' = use('desklamp_1')
    def use(self, object):
        ...

    # Clean an object with a receptacle.
    # For example, 'You clean soapbar_1 using sinkbasin_1.' = clean_with('soapbar_1
     ', 'sinkbasin_1')
    def clean_with(self, object, receptacle):
        ...

    # Heat an object with a receptacle.
    # For example, 'You heat tomato_1 using microwave_1.' = heat_with('tomato_1', '
     microwave_1')
    def heat_with(self, object, receptacle):
        ...

    # Cool an object with a receptacle.
    # For example, 'You cool pan_2 using fridge_1.' = cool_with('pan_2', 'fridge_1')
    def cool_with(self, object, receptacle):
        ...

# A useful function that you can use.
# Extract a list of object_ids with the specified object_name from the observation.
def get_object_with_id(observation, object_name):
    ...
```

[Response Instructions]
You will be given rules discovered by colleagues, summarizing game mechanisms and
 experiences of success and failure. Use these rules to guide your coding efforts.

```
Aim to understand and apply the principles behind the examples of these rules,
 adapting them to fit your specific scenarios within the simulated environment.

Output Format Instructions:
1. First, explain your understanding of the task and the current observations, then
 describe rules (with their IDs) that need to be considered and plan for the task.
 Then, write your code between "```python" and "```". No text should follow after
 the code block.
2. After receiving feedback, you should also explain your understanding of the
 current observations, including special (unexpected) formats or phenomena, mistakes
  in your previous code, and the reasons for errors. Then, describe rules (with
 their IDs) that need to be considered, plan for the task, and then write code.

Follow these instructions:
1. DO NOT USE an undefined function or attribute of the agent. Your code must be
 directly executable in the given environment. I won't implement any placeholders in
  your code for you.
2. Your code should be consistent with the code examples of the rules (please copy
 the code if there is no better modification!), making it easier for the builder
 agent to refine and develop new rules effectively.
3. In your code, you are encouraged to define helpful functions, which should be
 general and reusable in different scenes. The helper methods in the rules are
 already defined in the coding environment, so you can directly use them. If you don
 't need to modify them, don't redefine them.
4. After defining helper methods, your code should be divided into steps (marked
 with "[Step]" in comments) and write assertions at the end of each step to ensure
 that each step achieves its subgoal.
Follow these instructions. Don't give up or change the task.
```

Listing 4: Conclusion Prompt for Indirect Success

```
Please summarize the misunderstandings and mistakes you made, and then organize your
  code into a code block. You should copy the used parts from your previous code,
 including helper methods and steps. You can only modify the previously wrong step,
 and make sure you don't miss any detail! So that I can use it with similar but not
 the same scenarios.
```

Listing 5: Conclusion Prompt for Failure

```
You failed to complete the task. Now carefully review the trajectory of the executed
  actions and the corresponding observations, then identify the reasons for the
 failure. This reason is often your mistake or misunderstanding rather than the
 environment's error. By carefully comparing with applicable rules and examples,
 pinpoint where your code deviated from expected standards.

If there exist coding errors, specify the segments of the problematic code and
 elucidate how they contributed to the errors. If you want to write a correction for
  an error, you must double-check its plausibility! Additionally, determine whether
 your codes were in strict adherence to the relevant rules and examples. This
 reflection and documentation will serve as a reminder for completing future tasks.
```

## H.2   Examples for ALFWorld and MiniWob++

Listing 6: The Put Example for ALFWorld

```
You are in the middle of a room. Looking quickly around you, you see cabinet_4,
 cabinet_3, cabinet_2, cabinet_1, countertop_1, garbagecan_1, sinkbasin_2,
 sinkbasin_1, toilet_2, toilet_1.
Your task is to: find some spraybottle.

### Understanding of the observation: ...
### Rules to consider: ...
```

```
### Plan for the task: I need to get a list of receptacles, and go to search each
 receptacle until seeing a spraybottle.

Here is a code of my plan:
'''python
# Define a helper method to search receptacles for the target object
def find_object(agent, recep_to_check, object_name):
    for receptacle in recep_to_check:
        observation = agent.go_to(receptacle)
        # Check if we need to open the receptacle. If we do, open it.
        if 'closed' in observation:
            observation = agent.open(receptacle)
        # Check if the object is in/on the receptacle.
        if object_name in observation:
            object_ids = get_object_with_id(observation, object_name)
            return object_ids, receptacle
    return None, None

# [Step 1] Get a sorted list of receptacles and surfaces to check for a spraybottle.
  And use 'find_object' method to search
recep_to_check = ['cabinet_1', 'cabinet_2', 'cabinet_3', 'cabinet_4', 'countertop_1
 ', 'toilet_1', 'toilet_2', 'sinkbasin_1', 'sinkbasin_2', 'garbagecan_1']
object_ids, receptacle_with_spraybottle = find_object(agent, recep_to_check, '
 spraybottle')
assert object_ids is not None, f'Error in [Step 1]: There is no spraybottle in/on {
 recep_to_check}.'

# [Step 2] Take the spraybottle
found_spraybottle = object_ids[0]
observation = agent.take_from(found_spraybottle, receptacle_with_spraybottle)
assert agent.holding == found_spraybottle, f'Error in [Step 2]: I cannot take {
 found_spraybottle} from {receptacle}.'

# [Step 3] Go to a toilet and put the spraybottle on it
observation = agent.go_to('toilet_1')
# check if toilet_1 is closed. If so, open it.
if 'closed' in observation:
    observation = agent.open('toilet_1')
observation = agent.put_in_or_on(found_spraybottle, 'toilet_1')
'''
```

Listing 7: Initial Rules for ALFWorld Induced from Put Example

```
"rule_0":
{"rule": "At the initial observation of the environment, the agent can only observe
 receptacles, such as cabinet_1, countertop_1. The agent needs to go to the
 receptacle to view objects in or on it, even for open receptacles.",
"type": "Special Mechanism",
"example": "",
"validation_record": "Provided by User."},

"rule_1":
{"rule": "If there are multiple receptacles to be searched, the agent can write and
 use the 'find_object' method as shown in the example.",
"type": "Useful Helper Method",
"example": '''
    # Define helper method to find the object that is needed
    def find_object(agent, recep_to_check, object_name):
        for receptacle in recep_to_check:
            observation = agent.go_to(receptacle)
            # Check if we need to open the receptacle. If we do, open it.
            if 'closed' in observation:
                observation = agent.open(receptacle)
            # Check if the object is in/on the receptacle.
            if object_name in observation:
```

```
                    object_ids = get_object_with_id(observation, object_name)
                    return object_ids, receptacle
        return None, None
''',
"validation_record": "Essential helper method provided by User."},

"rule_2":
{"rule": "If the task is to put some object on some receptacle, first use '
 find_object' method in rule_1 to search all receptacles, take the object, then go
 to the target receptacle and put the object.",
"type": "Success Process",
"example": '''
    # For example, to put some spraybottle on toilet:
    # [Step 1] Use 'find_object' method in rule_1 to search all receptacles
    recep_to_check = ...
    object_ids, receptacle_with_spraybottle = find_object(agent, recep_to_check, '
     spraybottle')
    # [Step 2] Take the spraybottle
    # [Step 3] Go to a toilet and put the spraybottle on it
    # If the toilet is closed, open it first.
''',
"validation_record": "Provided by User."}
```

For MineWob++, the Planner is allowed to complete the task in steps: initially output segments of planned steps and pause to observe their outcomes to inform subsequent steps.

Listing 8: The `search-engine` Example for MineWob++

```
Your task is to Use the textbox to enter "Jerald" and press "Search", then find and
 click the 4th search result.
...(HTML)...

### Understanding of the observation: ...
### Rules to consider: ...
### Plan for the task: I need to click <input> textbox and type "Jerald". Then I
 need to click "Search" button and get the searching result. Finally, navigate to
 and click on the 4th search result.

Here is a code of my plan:
```python
# [Step 1] Find <input> textbox and type "Jerald".
textbox_xpath = "//*[@id='search-text']"
agent.click_xpath(textbox_xpath)
agent.type("Jerald")

# [Step 2] Click the search button
search_button_xpath = "//*[@id='search']"
html_string = agent.click_xpath(search_button_xpath)
# *Pause here*, waiting for the search to execute and for the results to be
 displayed.
```

### Feedbacks ###
...(some feedbacks)...
...(resulting HTML)...

### Understanding of the observation: ... Because one page only displays 3 search
 results, I need to turn to the next page for the 4th search result.
### Rules to consider: ...
### Plan for the task: ...
```python
# Define a helper method to turn to the next page.
def turn_to_next_page(agent):
  next_page_xpath = f"//*[@id='pagination']/li[@class='page-item next']"
  html_string = agent.click_xpath(next_page_xpath)
```

```
  return html_string

# [Step 3] Use 'turn_to_next_page' method to turn to the next page for the 4th
 search result and click the result
html_string = turn_to_next_page(agent)
# data-result start from 0
result_xpath = f"//*[@id='page-content']//a[@data-result='3']"
agent.click_xpath(result_xpath)
'''
```

Listing 9: Initial Rules for MineWob++ Induced from `search-engine` Example

```
'rule_0':
{"rule": "If the idx of the target search result exceeds 3 (the number of results
 per page), use 'turn_to_next_page' method as shown in the example.",
"type": "Useful Helper Method",
"example": '''
    # Turn to the next page.
    def turn_to_next_page(agent):
      next_page_xpath = f"//*[@id='pagination']/li[@class='page-item next']"
      html_string = agent.click_xpath(next_page_xpath)
      return html_string
''',
"validation_record": "Essential helper method provided by User."},

'rule_1': {
"rule": "If the task needs to use the search engine, first click <input> textbox,
 type the target string, and then click \"Search\" button. Next, use '
 turn_to_next_page' in rule_1 to turn the page if the target idx exceed 3 and click
 the target result.", "type": "Success Process",
"example": '''
    # For example, to search Jerald and click the 4th result:
    # [Step 1] Find <input> textbox and type "Jerald".
    # [Step 2] Click the search button.
    # [Step 3] Use 'turn_to_next_page' method in rule_1 to turn the page and click
     the target result.
''',
"validation_record": "Provided by User."}
```

Listing 10: The `enter-text` Example for MineWob++

```
Your task is to Enter "Ronda" into the text field and press Submit.
...(HTML)...
### Outputs ###

### Understanding of the observation: ...
### Rules to consider: ...
### Plan for the task: I should click the input box, type "Ronda" and click the
 submit button.

Here is a code of my plan:
```python
# [Step 1] Click the input box
inputbox_xpath = "//input[@id='tt']"
agent.click_xpath(inputbox_xpath)

# [Step 2] Type "Ronda" into the input box
agent.type("Ronda")

# [Step 3] Click the submit button
submit_button_xpath = "//button[@id='subbtn']"
agent.click_xpath(submit_button_xpath)
```
```

```
### Feedbacks ###
...(some feedbacks)...
obs_3: Act: agent.click_xpath("//button[@id='subbtn']"). Obs: Action done. The epoch
  is Done. Succeed: True.
```

## H.3  Prompts for Builder Agent

Listing 11: System Prompts for The Builder Agent

```
[Role]
You are observing a housekeeper agent as it codes and acts within a simulated
 environment (game). Your Role is to construct a manual of rules to not only assist
 the agent in completing tasks but also to do so with the least amount of attempts/
 errors. This requires recording and analyzing the experiences of the agent's
 successes and failures, and updating previous discoveries.

[Functions]
You will be presented with the current epoch's trajectory. The interaction in the
 trajectory includes the agent's analysis, execution code, and the resulting
 feedback.

You should use the following methods of rule_system to build and improve rules.

rule_system.write_rule(rule, type="", example="", validation_record="")
# Write down a new rule of the game you discovered.
# Parameters:
#   rule: a rule of the game you discovered.
#   type: the type of the rule, chosen from ["Special Phenomena/Mechanism", "
 Corrected Error", "Unresolved Error", "Useful Helper Method", "Success Process"].
 The "Corrected Error" can include misunderstandings and mistakes that have been
 corrected.
#   example: an example (or code) from the trajectory demonstrates this rule. You
 can add detailed information in the comment.
#   validation_record: your validation record on this rule, including the epoch IDs
 and rule IDs from which this rule is induced.

rule_system.update_rule(rule_id, rule="", type="", example="", validation_record="")
 ,
# Rewrite the attributes of an existing rule when you come up with a better
 understanding.
# Input only the attributes you want to rewrite.

rule_system.stop_generating()
# Description: stop generating rules from the current epoch.
# Use Case: When you believe that the trajectory of the current epoch is not needed
 or insufficient to derive any more new rules, you can call this function and wait
 for the next epoch's data. You should also call this function when you have updated
  all rules for the current epoch.

[Actions]
At each epoch, an agent is created in an environment, and the initial observation
 and target task are printed. The agent can only use the following action functions:

agent.go_to(receptacle) # Go to a receptacle and update the agent's location.
agent.open(receptacle) # Open a receptacle and observe its contents.
agent.close(receptacle) # Close a opened receptacle.
agent.take_from(object, receptacle) # Take an object from a receptacle if the agent
 is not holding anything.
agent.put_in_or_on(object, receptacle) # Put an object in or on a receptacle if the
 agent is holding it.
agent.use(object) # Use a lamp.
agent.clean_with(object, receptacle) # Clean an object with a receptacle.
agent.heat_with(object, receptacle) # Heat an object with a receptacle.
agent.cool_with(object, receptacle) # Cool an object with a receptacle.
```

```
get_object_with_id(observation, object_name) # Extracts a list of object_ids with
 the specified object_name from the observation.

[Response Instructions]
Upon receiving the current trajectory, you should first identify the case of the
 current trajectory's result and then build rules following the corresponding
 instructions.
```

Listing 12: Case Classify Prompts for Builder

```
Please analyze the scenario to identify the root cause of the observed mistakes and
 describe the existing rules related to the mistakes. Then check whether there
 exists a "Success Process" rule applicable to this type of task. Finally, determine
  whether the fault stems from:
- *Imperfect Rules*: the agent encounters unexpected phenomena that are not fully
 documented in the current rules, or the rules have not included the "Success
 Process" of this task type.
- *Imperfect Agent*: the rules fully document the "Success Process" and error
 reminders of such scenarios, but the agent fails to follow these rules meticulously
 .
Consider each step of the process carefully and conclude with either *Imperfect
 Rules* or *Imperfect Agent* based on your analysis.
```

Listing 13: Base Prompts for Builder

```
[Output Format Instructions]
Based on the current trajectory and your analysis, you should output the following
 things:
* Potential Rules: Describe your thoughts about potential rules based on the current
  trajectory. Depending on the results, you may need to check *Success Process*, *
 Helper Method*, *Corrected Error*, *Unresolved Error*, and other findings in
 sequence. Each potential rule needs to be clarified whether it is related to
 existing rules.
* Check Difference: Describe whether the potential rules target different phenomena.
* Check Existing Rules: Describe whether existing rules are conflicted or need
 updating.
* Codes: Finally, sequentially call the rule_system's functions within "```python"
 and "```".

[Detailed instructions]
**Maintain a maximum of 12 rules** Try to make each rule useful and non-repetitive,
 and insert new rules into closely related ones.
**Add Rules** Extract "Special Phenomena/Mechanism" rules when interactions appear
 counterintuitive, environment-specific, or when the agent expresses uncertainty
 about the environment mechanics (e.g., using "assume..." in the comment). Refrain
 from making speculative suggestions or guesses. Instead, conservatively document
 phenomena and the agent's valuable insights.
**Keep new rules targeted and precise.** Break down a large phenomenon or general
 strategy into targeted units as individual rules. These can later be upgraded or
 merged into a more general or larger rule.
**Write rules' scope** The time when the rule is triggered, or the task scope of the
  rule should be mentioned at the beginning of the rule.
**Avoiding overconfidence for new rules** Please acknowledge the need for further
 verification in your note.

**Update Rules** If an existing rule needs to be updated to include a new phenomenon
 , you should try to preserve the details of the existing content and preferably
 insert a categorial discussion or just insert new content into it (or its example).
  Especially, the rules of "Success Process" and "Useful Helper Method" type should
 retain their details. Then update the rule's validation_record after further
 verification or revision.
```

Listing 14: Case 1 Prompts for Builder

```
**Rules for Success** You should extract "Useful Helper Method" and "Success Process
 ":
* For each useful helper function identified: If it is not already included in a
 rule, create a rule of type "Useful Helper Method" and record its code unchanged in
  the rule's example section. If a method with similar functionality already exists
 in the rule, consider whether the rule needs to be updated.
* If the success process does not fall within the scope of an existing "Success
 Process" rule, faithfully document all steps (marked as "[Step]") in the successful
  code within a rule of type "Success Process", and document necessary codes and
 reminders in the rule's example; if the success process of the current task falls
 within the scope of the existing "Success Process" rule, consider whether the rule
 needs to be updated to incorporate the current roadmap.
```

Listing 15: Case 2 Prompts for Builder

```
**Rules for Success** You should extract "Useful Helper Method" and "Success Process
 ":
* For each useful helper function identified: If it is not already included in a
 rule, create a rule of type "Useful Helper Method" and record its code unchanged in
  the rule's example section. If a method with similar functionality already exists
 in the rule, consider whether the rule needs to be updated.
* If the success process does not fall within the scope of an existing "Success
 Process" rule, faithfully document all steps (marked as "[Step]") in the successful
  code within a rule of type "Success Process", and document necessary codes and
 reminders in the rule's example; if the success process of the current task falls
 within the scope of the existing "Success Process" rule, consider whether the rule
 needs to be updated to incorporate the current roadmap.

**Rules for Misstep** You should reflect on the main misstep to improve efficiency
 and log it into the "Corrected Error" type rule, including corrective code
 validated by the feedback (with the help of the agent's analysis and code, but its
 conclusion may not be correct and should be checked carefully).
```

Listing 16: Case 3 Prompts for Builder

```
**Rules for Success** You might need to update "Useful Helper Method" and "Success
 Process".
* For each useful helper method identified: If a method with similar functionality
 already exists in the rule, consider whether the rule needs to be updated.
* If the success process of the current task falls within the scope of the existing
 "Success Process" rule, consider whether you need to update the rule to include
 some tips or include important and specific code in its examples.

**Rules for Misstep** Identify existing rules that agents failed to follow and
 resulted in major mistakes. You should update the rule to emphasize some important
 points (you can add **...** at the part of the rule you want to emphasize) or to
 add error-prone points (perhaps added to the comments of the example code).
```

Listing 17: Case 4 Prompts for Builder

```
**Rules for Final Error** Based on your previous analysis and conclusion, summarize
 the final error that led to failure. You should write an "Unresolved Error" rule to
  record the error: in what situation, what the agent did, and what results were
 produced. So that they can serve as reminders for the agent in the future. Please
 don't rush to propose any definitive reasons or suggestions for the error; just
 record it.

The final error is unresolved and cannot be included in rules of other types than "
 Unresolved Error". As the task failed, you cannot write down any "Success Process"
 or "Useful Helper Method" rules.
```

Listing 18: Case 5 Prompts for Builder

```
**Rules for Misstep** Identify existing rules that agents failed to follow and
 resulted in major misstep. You should update the rule to emphasize some important
 points (you can add **...** at the part of the rule you want to emphasize) or to
 add error-prone points (perhaps added to the comments of the example code).

Remember that the rules of "Success Process" and "Useful Helper Method" type should
 retain their details.
```

## H.4  Prompts for Consolidator Agent

Listing 19: System Prompts for Consolidator

```
[Role]
You are observing a housekeeper agent as it codes and acts within a simulated
 environment (game). Your goal is to construct a manual of rules to assist the agent
  in completing various tasks in the environment. Your Role is to merge or delete
 previously found rules by analyzing the experiences of the agent.

[Functions]
You will be presented with the current found rules. The rules are extracted from
 many epochs' trajectories, in which each interaction includes the agent's analysis,
  execution code, and the resulting feedback.

A rule is represented with 'rule_id' and has the following attributes:
   - rule: the description of the rule, which begins with its use case or scope.
   - type: the type of the rule.
   - example: an example (or code) from the trajectory demonstrates this rule. You
    can add detailed information in the comment.
   - validation_record: your validation record on this rule, including the epoch IDs
     and rule IDs from which this rule is induced.

You should use the following methods of rule_system to delete and merge rules.

rule_system.update_rule(rule_id, rule="", type="", example="", validation_record="")
# Rewrite the attributes of an existing rule when you come up with a better
 understanding.
# Input only the attributes you want to rewrite.

rule_system.delete_rule(rule_id)
# Delete a existing rule with rule_id.
# **How to merge** To merge two existing rules, you can call rule_system.update_rule
  for one rule and then call rule_system.delete_rule to delete another rule.

rule_system.get_interactions(epoch_ids)
# Get the interaction history of previous epochs by their IDs.
# Use Case: You can use this tool to get the interactions from previous epochs (
 epoch starts from 0). You may need to check the validation_record of an existing
 rule to know which epochs to get.
# Parameters:
#   epoch_ids: a string containing the epoch IDs from previous epochs, separated by
 commas, e.g., epoch_0,epoch2.

rule_system.stop_generating()
# Description: stop generating rules from the current epoch.
# Use Case: You should call this function when you have finished updating all rules
 for the current epoch.

[Actions]
At each epoch, an agent is created in an environment. The agent can only use the
 following action functions in its code to interact with the environment:

agent.go_to(receptacle) # Go to a receptacle and update the agent's location.
agent.open(receptacle) # Open a receptacle and observe its contents.
```

```
agent.close(receptacle) # Close a opened receptacle.
agent.take_from(object, receptacle) # Take an object from a receptacle if the agent
 is not holding anything.
agent.put_in_or_on(object, receptacle) # Put an object in or on a receptacle if the
 agent is holding it.
agent.use(object) # Use a lamp.
agent.clean_with(object, receptacle) # Clean an object with a receptacle.
agent.heat_with(object, receptacle) # Heat an object with a receptacle.
agent.cool_with(object, receptacle) # Cool an object with a receptacle.
get_object_with_id(observation, object_name) # Extracts a list of object_ids with
 the specified object_name from the observation.

[Response Instructions]
Output Process:
After receiving the current rules, you should select potential rules to investigate
 and then delete or merge rules.

Detailed instructions:
**Maintain a maximum of 12 rules**
**Merge if addressed** If a "Success Process" rule can address the "Corrected Error"
  or "Unresolved Error" rule, you can consider merging these rules while retaining
 their details.

**Retain important details** The rules of "Success Process" and "Useful Helper
 Method" type should retain their details, and should not be deleted or easily
 refreshed by new updates. You cannot merge two rules of type "Success Process" or "
 Useful Helper Method"!
**Insertion is preferable** If a rule is updated to include the content of other
 rules, you should try to preserve the details of the existing content and
 preferably insert a categorial discussion or insert new content to it (or its
 example).
```

## H.5 Prompts for Formulator Agent

Listing 20: System Prompts for Formulator

```
[Role]
You are observing a housekeeper agent as it codes and acts within a simulated
 environment (game). Your goal is to construct a manual of rules to assist the agent
  in completing various tasks in the environment. Your role is to formulate a manual
  based on the found rules, including categorizing and summarizing related rules.

[Functions]
You will be presented with the current found rules. The rules are extracted from
 many epochs' trajectories, in which each interaction includes the agent's analysis,
  execution code, and the resulting feedback.

A rule is represented with 'rule_id' and has the following attributes:
   - rule: the description of the rule, which begins with its use case or scope.
   - type: the type of the rule.
   - example: an example (or code) from the trajectory demonstrates this rule. You
    can add detailed information in the comment.
   - validation_record: your validation record on this rule, including the epoch IDs
     and rule IDs from which this rule is induced.

[Actions]
At each epoch, an agent is created in an environment. The agent can only use the
 following action functions in its code to interact with the environment:

agent.go_to(receptacle) # Go to a receptacle and update the agent's location.
agent.open(receptacle) # Open a receptacle and observe its contents.
agent.close(receptacle) # Close a opened receptacle.
agent.take_from(object, receptacle) # Take an object from a receptacle if the agent
 is not holding anything.
```

```
agent.put_in_or_on(object, receptacle) # Put an object in or on a receptacle if the
 agent is holding it.
agent.use(object) # Use a lamp.
agent.clean_with(object, receptacle) # Clean an object with a receptacle.
agent.heat_with(object, receptacle) # Heat an object with a receptacle.
agent.cool_with(object, receptacle) # Cool an object with a receptacle.
get_object_with_id(observation, object_name) # Extracts a list of object_ids with
 the specified object_name from the observation.

[Response Instructions]
Output Process:
After receiving the current rules, you should output the following things:
* General Understandings: Describe your overall understanding of all rules and some
 specific rules.
* Category of Rules: Methodically analyze the connections between related rules,
 then cluster these rules, and propose category names for the clusters. Make sure
 each rule must belong to one and only one category!
* The Manual: Finally, sequentially write a structured manual within '''''markdown'
 and '''''. In the manual, you first describe the overview of all rules and then
 introduce each category of rules. In each category, you should list the rules and
 write rule_id within ** and **.

Detailed instructions:
1. Categorize rules based on their use cases and topics they target, not based on
 their "type".
2. If two "Success Process" rules follow the same critical success points or process
 , you can consider categorizing them into one category and propose a general
 strategy with the critical success points in the Introduction section of the
 category. But you don't have to do this if they don't follow the same critical
 success points.
3. To make the manual more accessible, please make the categories and rules appear
 in order from easy to difficult and from basic to complex.
```

