# OpenReview forum: "AutoManual: Constructing Instruction Manuals by LLM Agents via Interactive Environmental Learning"
_NeurIPS.cc/2024/Conference — NeurIPS 2024 poster_

### Official Review · Reviewer_3jfh · 2024-06-23

**Soundness:** 3
**Presentation:** 3
**Contribution:** 3
**Rating:** 6
**Confidence:** 4

**Summary:**

This paper introduces AutoManual, a framework enabling LLM agents to autonomously build understanding of new environments and generate instruction manuals through interactive learning. The system comprises three main components: a Planner agent generating code-based plans for environment interaction, a Builder agent updating a structured rule system based on observed trajectories, and a Formulator agent compiling rules into a comprehensive manual. To mitigate hallucinations in rule management, the authors implement a case-conditioned prompting strategy for the Builder. The framework is evaluated on the ALFWorld and MiniWoB++ benchmarks, demonstrating high success rates. AutoManual aims to address limitations of prior approaches such as path dependency and reliance on extensive human-provided instructions, potentially improving the adaptability and generalization of LLM agents in diverse environments.

**Strengths:**

Originality:

- Novel and creative framework for online rule learning and manual generation by LLM agents
- Innovative case-conditioned prompting strategy for rule management

Quality:

- Detailed examples and case studies demonstrating the approach
- Careful comparison to relevant baselines (e.g., ReAct, Reflexion, ExpeL)

Clarity:

- Clear explanation of the overall framework and key components
- Well-structured paper and easy to follow
- Detailed appendix with prompts, examples, and implementation details

Significance:

- Addresses an important challenge in building more general and adaptable LLM agents
- Demonstrates a path towards more autonomous learning in new environments

**Weaknesses:**

1. Heavy reliance on very capable LLMs like GPT-4, and dependency on oracle feedback during rule construction process limits broader applicability, do we have results by fine-tuning smaller models on the generated data?
2. Scalability to larger, more complex environments is unclear - the current approach puts all rules in LLM context. It’s not clear how easily the method can be generalized to other environments, for example, SWE-Bench.
3. Limited evaluation of the quality and usefulness of the generated manuals for humans. Lack of theoretical analysis on the rule learning process.

**Questions:**

1. How is the initial rules and examples selected and how sensitive is the performance to the quality of the initial rules/examples provided? How this initialization needs to be changed when we switch to different environments?
2. In Section 3.2 Skill and Reflection library construction, the most similar task is retrieved from the skill library when new task comes, how is the similarity between task been defined?
3. In 3.3 Rule types definition, how does the 6 rule types come up?
4. Table 2 shows the success rate of LLM agents on 9 task types with feedback, what are those feedback, and is the feedback rely on an oracle who know the correct answer/solution for the given query?

**Limitations:**

The authors provide a reasonable discussion of limitations, including reliance on GPT-4 and potential scaling challenges. However, more concrete mitigation strategies are missing. They also touch on broader impacts, noting both positive potential (e.g., improved safety, worker training) and risks (e.g., unpredictable behaviors).

Side note: should the limitation and broader impact discussions be included within the main 9-page scope rather than in the Appendix?

---

> ### Author Rebuttal · Authors · 2024-08-04
>
> 1- **Fine-tune Smaller Models:** We are unsure about the "generated data" referred to by the reviewer. Is this data generated through interactions between GPT-4 and the environment to create successful trajectories? This also seems to require the use of very capable LLMs. It is important to note that the goal of our work is to enable LLM Agents to quickly and autonomously adapt to unfamiliar new environments. These environments do not have sufficient high-quality human or generated data for fine-tuning smaller models. Our method requires only 36 turns of interactions with the environment to build a Manual, whereas accumulating enough data to fine-tune an LLM would require an order of magnitude more interactions.
>
> 2- **Scalability to Larger, More Complex Environments:** We currently place all rules in the context to ensure that the Planner can access all the rules, thus eliminating the impact of rule retrieval on our experimental outcomes. However, for more complex and larger environments, leveraging technologies like Retrieval-Augmented Generation (RAG) is indeed necessary to effectively address the challenges. To demonstrate our method's potential in handling complex environments, we conducted experiments on the Reddit site of WebArena. WebArena [1] is a high-difficulty benchmark simulating real website interactions. In this environment, we set a maximum of 15 rules for AutoManual. We used AutoManual paired with GPT-4-turbo to build and formulate tasks in the Reddit site environment, subsequently testing on 106 tasks. The test results and comparisons with other works are shown in the table below. We provide the code and trajectories at [our comment below]. We will also include these experimental results in our paper.
>
> Notice: Although SWE-Bench is also a challenging benchmark, its interactivity is limited. Therefore, we chose a more interactive environment, WebArena, which aligns better with the "Interactive Environment" in our paper's title.
>
> Table: Test on WebArena (Reddit) using GPT-4-turbo
> | Methods       | Examples | Success Rate(%) |
> | ------------- | -------- | --------------- |
> | ReAct [1]     | 2        | 6.0             |
> | AutoGuide [2] | 19       | 43.7            |
> | SteP [3]      | 14       | 55.0            |
> | AutoManual    | 1        | 65.1            |
>
> 3- **Sensitivity Analysis of AutoManual on Examples and Initial Rules:** In Appendix F.2, we analyze the sensitivity of AutoManual to the example and initial rules. We provided AutoManual with an example of the "enter-text" task from MiniWob++, which is the simplest task type in this suite, along with the corresponding initial rules. Our findings suggest that while simpler examples and initial rules do influence the outcomes, the impact is minimal.
>
> **Strategy for Generating Initial Rules and Examples:** Generally, for a new environment, we first select one task to serve as the example task, with no specific criteria for selection (In the ALFWorld experiments, we chose the simplest "put" task and achieved significant results). Then, a human operator generates a trajectory that completes the task, thus providing the necessary example. Initial rules typically comprise a "special mechanism" and a "success process" type rule, which involves considering any counterintuitive aspects of the environment and summarizing strategies to accomplish the example task.
>
> 4- **Task Similarity:** We follow the vector similarity approach used by Voyager and Adaplanner for skill retrieval, utilizing OpenAI's embedding model to process task names into embeddings. These embeddings serve as queries and keys, with the vector similarity from the FAISS library used to retrieve the most similar skill.
>
> 5- **How The Six Rule Types Were Created:** In "Rule System" of Section 3.3, we have mentioned the selection of the six rule types used in our rule system. Specifically, we believe that knowledge about the environment necessitates recording the transition functions, $T(o'|o, a)$ for observable states and $T(s'|s, a)$ for system states, as well as understanding how trajectories $\tau_{\rho}$ lead to outcomes of r=0 or r=1. These insights correspond to the task types: "Special Phenomenon" for $T(o'|o, a)$, "Special Mechanism" for $T(s'|s, a)$, and "Success Process" for $r(\tau_{\rho})=1$, "Unsolved Error" $r(\tau_{\rho})=0$. Additionally, the analysis of task segments within trajectories helps in achieving overall task success by identifying successful or failed sub-goals. This leads to the rule types "Useful Helper Method" and "Corrected Error."
>
> 6- The terminology "9 task types with feedback" and "44 task types without feedback" used in our paper follow previous works, Adaplanner. The "44 task types without feedback" denotes tasks where the HTML of the webpage remains static and does not change in response to the actions performed. For example, in the "click-button-sequence" task type, the user is required to click buttons One and Two, yet no observable changes in the HTML occur during these actions; the reward is provided only at the end of the task. Conversely, the "9 task types with feedback" refer to tasks where the webpage reacts to the actions taken. An illustrative example is the "tic-tac-toe" task, where after the agent makes a move, the webpage (the opponent) responds by making its move based on the agent's actions. This makes the "9 task types with feedback" more challenging as they represent more complex tasks, and the agent does not have full visibility of the webpage's content at the start.
>
> [1] Zhou, Shuyan, et al. "Webarena: A realistic web environment for building autonomous agents." arXiv preprint arXiv:2307.13854 (2023).
>
> [2] Fu, Yao, et al. "Autoguide: Automated generation and selection of state-aware guidelines for large language model agents." arXiv preprint arXiv:2403.08978 (2024).
>
> [3] Sodhi, Paloma et al. "Step: Stacked llm policies for web actions." arXiv preprint arXiv:2310.03720 (2024).

---

> ### Author Response · Authors · 2024-08-05
> **Code for "AutoManual on WebArena (Reddit)"**
>
> We provide the code and trajectories at https://anonymous.4open.science/r/automanual-3860.

---

> > ### Comment · Reviewer_3jfh · 2024-08-07
> >
> > The traces of the test results requires email and access request.

---

> > > ### Author Response · Authors · 2024-08-08
> > >
> > > We have updated its access permissions, and you can download traces at https://send.cm/d/1DdXT

---

> > > > ### Comment · Reviewer_3jfh · 2024-08-08
> > > >
> > > > In order to reproduce Alfworld, the following packages are required to be adjusted:
> > > > ```
> > > > pip install Werkzeug==2.2.2
> > > > pip install langchain-community langchain-core langchain_openai
> > > > ```
> > > >
> > > > I appreciate the authors' rebuttal, as they have addressed my questions about the framework and method. That being said, I would recommend including the above clarification in the final version. Meanwhile, I maintain a different perspective on the interaction complexity of tasks such as SWE-Bench and the coding tasks in AgentBench. In my view, these environments involve significant interaction, particularly in the form of responding to automatically generated feedback, such as stdout.
> > > >
> > > > A critical characteristics of a LLM agent is the agent's ability to interpret and act upon system feedback in different modalities. For instance, in the coding environment, when an agent misuses a command, the terminal often suggests using `<command> --help` for more information. However, many off-the-shelf agents struggle to leverage these suggestions, instead opting to generate new responses without gathering the available additional information.
> > > >
> > > > I believe the mechanism proposed in this paper has the potential to address these information retrieval behaviors under certain conditions. This is precisely why I strongly recommend expanding the evaluation to include environments with different modalities. This would demonstrate that the AutoManual's success extends beyond navigation-like tasks (such as household household navigation in ALFWorld or web navigation in MiniWoB++ and WebArena) to a broader range of real agent tasks.
> > > > Testing the proposed method in these varied environments would provide compelling evidence of its versatility and effectiveness across different types of agent interactions.
> > > >
> > > > Given these considerations, I maintain my current score for the paper.

---

### Official Review · Reviewer_HWDR · 2024-07-12

**Soundness:** 3
**Presentation:** 4
**Contribution:** 3
**Rating:** 8
**Confidence:** 3

**Summary:**

LLM agents show promise in many environments but require expert prompts in specific environments to perform well. This paper introduces the AutoManual framework which autonomously builds a 'manual' of rules through interactions to help adapt to new environments. The framework consists of 3 agents: the Planner which generates code to carry out a plan, the Builder which updates rules, and a Formulator which consolidates the rules into a manual. The Builder updates rules through a original case-conditioned prompting strategy to mitigate hallucinations. AutoManual is evaluated on 2 benchmarks: AlfWorld and MiniWoB++ and achieves high performance.

**Strengths:**

$\textbf{Originality}$: AutoManual breaks apart the knowledge acquisition problem into 3 agents, one of which uses a case conditioned prompting strategy, which breaks down rule updates into a decision tree that simplifies LLM responses, mitigating hallucinations.

$\textbf{Quality}$: AutoManual is evaluated on 2 benchmarks against popular baselines and related knowledge acquisition works and provides insightful ablations into its components. High quality.

$\textbf{Clarity}$: Plenty of helpful figures and very clear approach.

$\textbf{Significance}$: Autonomously collecting knowledge to formulate an instruction manual simplifies human efforts when developing prompts for LLM policy approaches.

**Weaknesses:**

This paper is pretty strong already so there are only 2 things I wanted to point out that could make it stronger.

1. The ablations are very helpful in justifying design decisions but the table looks odd with seemingly random kinds of ablations. I understand if there is a concern with resource-constraints running all 16 combinations of ablations but why this set of 7 ablations?
2. Temperature 0 for OpenAI is typically a red flag to me that performance is brittle since there are no guarantees of determinism [1]. Despite experiments being run 3 times and averaged, I've experienced varying performance across different days. I'd be interested to see if a subset of the results remains relatively the same.

[1] https://community.openai.com/t/why-the-api-output-is-inconsistent-even-after-the-temperature-is-set-to-0/329541

**Questions:**

Suggestions
- Provide an explanation/justification for the incomplete ablation table in the paper
- Provide a subset of results (e.g. a single column) to verify performance isn't brittle due to temperature 0

Questions
-  Manuals are constructed by sampling specific task types in both environments. I am curious whether you experimented with constructing manuals with limited task types and testing on unseen task types. In other words, can the manuals be used to perform other tasks (e.g. for AlfWorld only constructing a manual out of Put but testing on Put Two)
- Despite the case conditioned prompting to mitigate hallucinations for the Builder, the Planner can ignore these rules as noted in the limitations. Is this the most common failure case and, if so, what were the practical solutions that were taken to make the Planner pay attention to the Builder's rules?
- As a follow-up, AlfWorld and MiniWoB++ are environments where there is likely overlap in the LLM's training data and the benchmarks (home environment and web-based environments); however, how would AutoManual perform in environments that may not align with the LLMs priors (e.g, Atari games or Sokoban)

**Limitations:**

The limitations section is transparent and the broader impacts section addresses societal impact.

---

> ### Author Rebuttal · Authors · 2024-08-02
>
> 1- **Significance of Selected Ablations**:
>
> Each row in the ablation study table in Section 4.2 was carefully chosen to demonstrate the contribution of different components of AutoManual to the overall results. Specifically:
>    - The first row, "Planner only," serves as our baseline, testing only our Planner Agent without any building process.
>    - The second row, "Planner + Lib.," involves building only the Skill & Reflect Library during the building process, highlighting that merely recording skills and reflections without constructing rules is insufficient.
>    - The third row shows the "Offline Learning" version of AutoManual, illustrating that using offline trajectories rather than online learning significantly reduces effectiveness, thereby supporting our claims made earlier in the paper.
>    - Rows four to six respectively demonstrate the impacts of removing the Skill & Reflect Lib., case-conditional prompting, and manual formulation components from AutoManual, underlining the necessity of these elements for optimal performance.
>    - The final row represents the full version of AutoManual, achieving the best results.
>
>    These meanings also correspond to the comparison of each paragraph in Section 4.2. Other combinations have no special meaning or comparative significance, so they are not included in the results. We will make the significance of each row more explicit in the revised manuscript.
>
> |                       Meaning                       | Online | Skill&Reflect Lib. | Case Prompt | Formulation | Success Rate (%) |
> | :-------------------------------------------------: | :----: | :----------------: | :---------: | :---------: | :--------------: |
> |                    Planner only                     |        |                    |             |             |       77.6       |
> |                    Planner+Lib.                     |        |         √          |             |             |       88.1       |
> |                  Offline Learning                   |        |         √          |      √      |      √      |       90.7       |
> |                AutoManual w.o. Lib.                 |   √    |                    |      √      |      √      |       89.5       |
> | AutoManual w.o. case-conditional prompting strategy |   √    |         √          |             |      √      |       93.8       |
> |             AutoManual w.o. formulation             |   √    |         √          |      √      |             |       96.5       |
> |                     AutoManual                      |   √    |         √          |      √      |      √      |       97.4       |
>
> 2- **GPT's Temperature**:
>
> We also noted that setting the temperature to 0 for GPT does not always produce identical results, but the impact on our experiments is minimal. This is evident from the standard deviation, approximately 2, shown in Figure 4. In fact, the three runs in each result were not completed on the same day; the experiments often differed by many days, and then the average was taken. To confirm the stability of our AutoManual, we have conducted additional experiments with varying temperatures and 3 runs below.
>
>    Table: Additional Experimental Results of AutoManual on ALFWorld
>
> | Method                  | Temperature | Runs | Success Rate（%） |
> | ----------------------- | ----------- | ---- | --------------- |
> | AutoManual（gpt-4-turbo） | 0           | 1    | 96.3            |
> | AutoManual（gpt-4-turbo） | 0           | 2    | 98.1            |
> | AutoManual（gpt-4-turbo） | 0           | 3    | 96.7            |
> | AutoManual（gpt-4-turbo） | 0.5         | 1    | 95.6            |
> | AutoManual（gpt-4-turbo） | 0.5         | 2    | 96.3            |
> | AutoManual（gpt-4-turbo） | 0.5         | 3    | 97.6            |
>
> 3- **Task Generalization in AutoManual**: You raised an excellent point regarding task generalization. But we have to say that each task type has its own unique solution, so the Manual built on some tasks may not have a high test result on the remaining tasks. For example, AutoManual only constructs a manual out of Put, but when testing on Put Two, it will require specific strategies and an understanding of the environment: For Put Two, the agent needs to know that it cannot take two things at the same time, and should employ a strategy of taking one and putting one, etc. However, for some similar task types, AutoManual does have task generalization. For instance, removing one of the clean, heat, and cool task types during building and testing across all tasks showed a success rate nearly identical to 97.6%. This is because strategies for these tasks can be easily generalized from one to another.
>
> 4- **The Planner Can Ignore Rules**: This issue, as noted in the Limitations of the Appendix, is indeed recurring and challenging. We may consider integrating an additional validator agent to re-confirm whether the Planner adheres to the set rules or commits errors, using techniques like self-refine, but this would significantly increase token consumption. We leave this challenge to future research.
>
> 5- **Other Environments**: It is important to note that almost no environment is both practical and unrelated to GPT’s training data. For famous games like Atari or Sokoban, relevant knowledge is likely included in GPT's training dataset. Given that Atari's [Instruction Manual](https://atariage.com/system_items.php?SystemID=2600&itemTypeID=MANUAL) is a public document, GPT has likely seen and memorized the strategies for playing these games. In addition, even though home and web-based environments might overlap with GPT's training materials, our experiments show that relying solely on GPT's inherent knowledge (Planner Only) achieved only a 77.6% success rate, indicating the insufficiency of intrinsic knowledge for solving tasks in these environments. Our AutoManual method effectively supplements environmental knowledge through trial and error.

---

> > ### Comment · Reviewer_HWDR · 2024-08-09
> > **Response to Authors**
> >
> > Thank you for the response. I appreciate running the temperature experiment - I am happy to see that the results are relatively robust and am excited to see this applied to other domains in the future! I have increased my score.

---

### Official Review · Reviewer_azPd · 2024-07-13

**Soundness:** 2
**Presentation:** 3
**Contribution:** 3
**Rating:** 6
**Confidence:** 4

**Summary:**

The authors propose a new approach to interactive environment interactions by LLMs which they dub AutoManual. The approach takes an LLM and uses it in several different roles, differentiated by what context is included in their prompts, and some number of saved rules. The fundamental idea of the approach is to optimize over the saved rules in order to improve the performance. The authors demonstrate that AutoManual can perform well with a single human-example in both ALFWorld and MiniWob++ compared to baseline approaches and they demonstrate the importance of its various components through an ablation study.

Edit: I have updated my review based on the authors' response.

**Strengths:**

The paper proposes what appears to be a useful framework for applying LLMs to decision making tasks. Clear arguments are made for the various components of the framework, which are all fully described in the appendices. The ablation study meaningfully demonstrates the additive value of the various components, and the comparison to baselines demonstrates strong performance.

**Weaknesses:**

I have one concern with the paper and a few minor concerns.

My major concern has to do with the reported performance of AdaPlanner. The authors acknowledge that they could not replicate the performance but justified this that AdaPlanner's format requirements were too strict. The original authors, Sun et al. identified that the issue with 3.5-turbo (and one would assume later models) was their tendency to hallucinate. If AutoManual is robust to these hallucinations that would be excellent, but it would make sense to include an AdaPlanner implementation with gpt-3 given that is the stated preferred models by AdaPlanner's authors. Further, the results with gpt-3 appear to meet or exceed the performance of AutoManual. This is by far my largest concern with the current paper draft and I'd strongly encourage the authors to include an AdaPlanner gpt-3 baseline or explain their reasoning not to.

In terms of minor concerns, there are a number of claims that are somewhat overblown in the current paper text such as "to enhance their readability and global understanding" when no results demonstrate improved readability or global understanding. Similarly, the authors claim the approach is "user-friendly" and "like a teacher" in the same paragraph, given that the authors do not include any human subject study or other user tests these are unsupported in the current draft. "making it grounded and well-understood" would be yet another unsupported claim. Another minor concern is that some implementation details and associated design decisions are not given, such as not specifying the value of Nmax used for any of the experiments. Finally, the language is somewhat off in certain parts such as "the function should be called"->"this function should be called", "like a teacher conveys"->"like how a teacher conveys", and so on.

**Questions:**

1. Why didn't the authors include a gpt-3 baseline for AdaPlanner?
2. Do the authors feel AutoManual handles hallucinations better? If so, why?
3. Can the authors given the value of N_max and other hyperparameters missing from the paper?

**Limitations:**

Yes the authors have adequately addressed the limitations.

---

> ### Author Rebuttal · Authors · 2024-08-01
>
> ## Regarding the Exclusion of AdaPlanner's GPT-3 Results:
>
> 1- **Reproducibility Issues with AdaPlanner's GPT-3**: The high success rate claimed by AdaPlanner's GPT-3 is nearly impossible to replicate. We utilized the publicly available code from AdaPlanner and tested it multiple times using OpenAI's davinci-002 and gpt-3.5-turbo-instruct models, only achieving success rates between 50%-70%. During our trials, we identified and fixed several formatting bugs in the code, yet the highest success rate plateaued at 76%. Additionally, others have also reported difficulties in replicating these results in the issues section on AdaPlanner's GitHub [see our comments below]. We invite reviewers to personally test the code to see the reproducibility of the results.
>
> _Note: OpenAI discontinued text-davinci-002 and text-davinci-003 models as of January 4, 2024, leaving davinci-002 as a substitute and recommending users to switch to gpt-3.5-turbo-instruct which is purported to have equal or enhanced capabilities._
>
> 2- **Lack of Implementation Details for "Skill Filtering"**: The AdaPlanner paper discusses a crucial "Skill Filtering" stage in Section 3.3, yet we found no implementation of this in the provided code, nor is it detailed in the paper. According to the paper, this filtering boosts success rates significantly, as shown in Figure 4. In addition, implementing such filtering would require testing a substantial subset, if not all, of the test set multiple times. This frequent validation using the test set could unfairly benefit AdaPlanner over other methods that do not use the test set in such a manner.
>
> 3- **GPT-3 Tends to Copy Curated Examples**: Adaplanner meticulously designed its human examples to the extent that they only require replication to fulfill the task. Adaplanner provided one example for each of the six task types in ALFWorld, each example comprehensively offering the code needed to solve the respective task. This resulted in a scenario where, faced with a specific task type, one only needs to change the item names and target container names in the human example to produce a working code solution. To validate this point, we conducted an experiment:  For a new task, we replaced the item and target names in the human example with those from the new task description, and ran the resulting solution function. We discovered that this direct copying approach (simply replacing names) achieved a success rate as high as 70%. Furthermore, we found that 30% of the failures were mainly due to errors from the sub-function `ask_LLM()` rather than issues with the solution itself. The code and traces for this experiment are available at [our comments below].
>
> Given such potent human examples, it becomes less about the LLM agent's planning abilities and more about who can best copy and make minor modifications. Compared to the high innovativeness of gpt-3.5-turbo, gpt-3 tends to replicate text from examples. This highlights the truth behind the authors' statement that "gpt-3.5-turbo (and one would assume later models) was their tendency to hallucinate": the newer models are more inclined to generate answers that differ in form from the perfect human examples. However, we believe that simply replicating these artificially designed solutions does not serve as a fair comparison. By contrast, our AutoManual system does not face this issue, as it was only provided with a single human example for the most basic task.
>
> For these reasons, we have excluded AdaPlanner GPT-3's extraordinary result from our comparative analysis. We believe it is our duty to provide **fair and reproducible** experimental results as a standard for comparison for our readers and for future research.
>
> ## On Exaggerated Claims:
> We acknowledge that we did not conduct human subject testing on "readability or global understanding" for AutoManual. However, the enhanced readability of sorted and categorized Markdown Manual text over a list of unordered rules is both intuitive and supported by established research [1]. We have provided the final generated Markdown Manual in the appendix (Figures 1 and 2) so reviewers can assess its organization and readability themselves. Proper categorization and arrangement not only improve readability but are also known to enhance LLM performance, aligning with human experience and empirical evidence [2].
>
> We appreciate your feedback, and we will adjust our language to moderate the claims made.
>
> ## Additional Questions:
> In Appendix D, we mentioned our choice of `N_max` as 12 to avoid overly long context issues. This section also includes detailed descriptions of our hyperparameters. We will make these more prominent.
>
> [1]: Beck, I. L., McKeown, M. G., Sinatra, G. M., & Loxterman, J. A. (1991). Revising social studies text from a text-processing perspective: Evidence of improved comprehensibility. Reading Research Quarterly, 27, 25 1-276.
>
> [2]: [How I Won Singapore’s GPT-4 Prompt Engineering Competition](https://towardsdatascience.com/how-i-won-singapores-gpt-4-prompt-engineering-competition-34c195a93d41)

---

> > ### Comment · Reviewer_azPd · 2024-08-08
> > **Re: Rebuttal by Authors**
> >
> > Thanks to the authors' for their detailed response! My major concern around Adaplanner has been addressed. I would suggest including a truncated version of that in the next version of the paper. As such, I have increased my score.
> >
> > I disagree with the characterization that the claims around human judgement are intuitive. I also do not agree that reviewers' subjective opinions would be sufficient to support these claims from the provided examples. I believe that arguments around human response to any system or system output require a human subject study for support.

---

> ### Author Response · Authors · 2024-08-05
> **Publicly available code of AdaPlanner; The issue with reproducing AdaPlanner; Code for "3- GPT-3 Tends to Copy Curated Examples"**
>
> The publicly available code of AdaPlanner [https://github.com/haotiansun14/AdaPlanner]
>
> The issue of reproducibility on AdaPlanner's GitHub [https://github.com/haotiansun14/AdaPlanner/issues/2]
>
> The code and traces for the experiment in "3- **GPT-3 Tends to Copy Curated Examples**" are available at [https://anonymous.4open.science/r/adaplanner_alfworld-F4B7].

---

### Official Review · Reviewer_zSsL · 2024-07-14

**Soundness:** 3
**Presentation:** 2
**Contribution:** 2
**Rating:** 5
**Confidence:** 4

**Summary:**

The authors propose AutoManual, a method for improving LLM agents by collecting and updating rules online via rule-assisted interactions in the environment, where code form planning is used for the interactions. In addition to the management of rules, AutoManual also maintains skill and reflection libraries per task type, where successful trajectories form the skill libraries and failed trajectories form the reflection libraries. The authors empirically show that AutoManual outperforms the compared baselines in ALFWorld and MiniWoB++ environments.

**Post-Rebuttal/Discussion**: While I appreciate the detailed author response, I don't think it fully addresses the major concerns I had. For example, by "maintaining the skill and reflection libraries per task type (L166)", I meant there exist explicitly and discretely isolated skill and reflection libraries, one for each task type. This design is made by the humans, not the LLMs. It may be doable for ALFWorld, which is comprised of clearly separable task types, but it would require more advancement for more complex environments. Therefore, I maintain my original score.

**Strengths:**

- The proposed method, AutoManual, alternates between exploiting and updating the set of rules, which is sound as the rules also get updated online based on the relevant experience.
- The authors show that the proposed method exhibits strong performance in ALFWorld and MiniWoB++, by comparing it empirically with multiple baselines. They also provide a fair amount of analyses including the ablation studies, which help to understand the proposed method in empirical aspects.

**Weaknesses:**

- While the presented empirical evaluations mainly compare the number of examples used as a measure of the amount of domain or human knowledge provided to the methods, such knowledge can be injected in different forms, e.g., input prompts or the exposure of LLMs such as GPT to the benchmarks and corresponding execution codes during their training. But another point I notice is that, this work injects such knowledge in a more explicit or discrete form by maintaining the skill and reflection libraries per task type (L166), which contributes most to the performance according to Table 3.
- Over-claiming statements need to be clarified or toned down. For instance, the Abstract claims that LLM agents "typically require elaborate design and expert prompts" in specific domains and suggests AutoManual as a solution. However, it appears that AutoManual leverages human-designed prompts for each domain as well as domain knowledge for design (as mentioned above).
- The experimental settings for the GPT-3.5-turbo results may not be entirely fair. According to Table 1 from Appendix D, the "GPT-3.5-turbo" runs with all the baselines are actually based on GPT-3.5-turbo except for the Builder component for ExpeL. It means that the Actor and Self-reflection for offline trajectories for ExpeL are also equipped with GPT-3.5-turbo, unlike Planner and Builder for the building stage for AutoManual.
- The manuscript can be improved in terms of presentation. especially the choice of technical vs non-technical terms. For instance, "free-form code" appears multiple times, but I couldn't find clarifications on what exactly "*free-form* code" is and whether (and how) it is different from just "code", which I assume is not. I have a similar concern about "natural programming capabilities" (L141).

**Questions:**

* Are the "initial rules" made or selected by humans or fully automatically derived by Builder?
* Please take a look at the Weaknesses section above.

**Limitations:**

The authors discussed the fair limitations in the Limitations section.

---

> ### Author Rebuttal · Authors · 2024-07-31
>
> 1- **Clarification on Environment-Specific Information:**
>
>    First, the environment-specific information provided by humans can be categorized into four types:
>
>    - (i) Brief description of the environment and roles (about 25 words)
>    - (ii) Actions that the agent can perform
>    - (iii) Examples of humans completing the task
>    - (iv) Potential environmental knowledge embedded during LLM training
>
>    Among these, Types (i) and (ii) are fundamental information for the agent to perform tasks in the environment, and can easily and justifiably be included in the prompts. Type (iv) is challenging to control and eliminate, but in our experiments, all methods used the same LLM and the same types (i) and (ii) prompts to balance the influence of this knowledge. Type (iii), the examples, are the most labor-intensive, each exceeding 200 words (ReAct and Reflexion require 12 such examples). To further illustrate the significance of examples in previous methods, we demonstrate in the table below that the performance of Reflexion (GPT-4-turbo) with only two 'put' task examples drops from 85.9% to 73.1%. Our AutoManual system requires just one human-provided example (and derived initial rules).
>
>    Additionally, as stated in Section 3.2, all content in the skill and reflection libraries is proposed and updated by the LLM Agent itself, not manually injected, which differs from "the examples."
>
> 2- **Reduction of Human-designed Prompts:**
>
>    As previously mentioned, types (i) and (ii) of the LLM Agent's input prompt are fundamental environmental information, while type (iv) is inherently uncontrollable during training. The main human effort is concentrated on type (iii), with prior works requiring multiple human-written environment-specific examples (each over 200 words), whereas AutoManual needs just one from humans. Furthermore, some previous Agents, such as Generative Agents [14] and Voyager [23], were specifically designed for their respective gaming environments within their frameworks, but we do not tailor our framework to any particular application environment. Therefore, we claimed that AutoManual significantly reduces the need for elaborate design and domain knowledge.
>
>   We appreciate your feedback and will tone down the claims in the Abstract accordingly.
>
> 3- **Experimental Fairness with Expel**:
>
> Thank you for highlighting this. We followed the original settings of the Expel paper in our experiments with Expel (GPT-3.5-turbo), which indeed might not have been entirely fair compared to our methods (GPT-3.5-turbo). We have added the experiment settings you mentioned below, where both Builder and Offline Trajectory’s Actor and Reflector use GPT-4-turbo, while Task Inference uses GPT-3.5-turbo. The results show that although the new Expel setup with GPT-4-turbo improves performance, it still does not match AutoManual's results: 67.2% vs. 86.2%. As discussed in our paper, this is due to Expel’s reliance on offline learning rather than our online learning, its rule management is rudimentary, etc.
>
> 4- **Clarification of Terminology**:
>
> The term "free-form code" is defined in Section 3.2, paragraph two, as: "This form simplifies planning by only generating code necessary for the current environmental situation and feedback without the overhead of integrating previously executed code." We use "free-form code" to distinguish from the "function-form code" used by Voyager and AdaPlanner. They require the output code to be a Python function, while our approach imposes no such restriction.
>
> The term "natural programming capabilities" is described in the paper CodeAct [24] as "Code data is widely used in pre-training today’s LLMs. These models are already familiar with structured programming languages." We appreciate your suggestion and will clarify these definitions further.
>
> 5- **Initial Rules**:
>
> 'Initial rules' are extracted by humans from the single provided example, demonstrating how rules should be extracted to the Agent. In Appendix F.2, we investigated whether the provided initial rules and that one example impact results, finding AutoManual robust to such variations.
>
> **Table: Additional Results**
> | Methods                    | Examples | Put  | Clean | Heat | Cool | Examine | Put two | ALL  |
> | -------------------------- | -------- | ---- | ----- | ---- | ---- | ------- | ------- | ---- |
> | Reflexion (gpt-4-turbo) | 2 (put)   | 88.9 | 90.8  | 66.7 | 88.9 | 55.6    | 33.3    | 73.1 |
> | ExpeL (gpt-3.5-turbo) (updated setting) | 12       | 80.6 | 62.4  | 71.0 | 71.2 | 64.8    | 51.0    | 67.2 |

---

> > ### Comment · Area_Chair_y8bc · 2024-08-14
> > **Reviewer Engagement**
> >
> > Dear Reviewer,
> >
> > Please remember to engage with the authors.
> >
> > Thanks!
> >
> > AC

---

### Author Rebuttal · Authors · 2024-08-05

We have provided detailed rebuttals for each reviewer's Weaknesses and Questions. We kindly ask reviewers to review the respective rebuttals, and if you find that our rebuttals have addressed your concerns, please consider raising your scores. Thank you very much for your patience.

---

### Decision · Program_Chairs · 2024-09-25

**Decision:**

Accept (poster)

**Comment:**

This paper introduces AutoManual, which is a framework built upon LLMs to facilitate planning using three agents; Planner, Formulator, and Builder. This planning framework automatically improves with experience by codifying (Formulator) and improving/revising (Builder) a set of planning rules which can then be used by the Planner to accomplish a goal. AutoManual is validated on two benchmarks against various baselines.

The reviewers were mixed but not negative. Reviewers shared concerns that the > 50 tasks evaluated were 'homogeneous' in nature, which called into question how generalizable the approach was. There were also remaining questions about how much sensitivity there was to prompt engineering due to the temperature = 0.

The reviewers agreed that there was originality, positive results (including ablations), and sound results. While concerns remain about generalizability, I am inclined to view this paper on the positive side due to the careful way in which the approach was evaluated, the positive results, and the contribution that this paper makes towards open-ended learning with LLMs for planning, which is a difficult problem for LLMs.